# Endothelial Pim3 kinase protects the vascular barrier during lung metastasis

Niina M. Santio[1,7], Keerthana Ganesh[1,7], Pihla P. Kaipainen[1], Aleksi Halme [1], Fatemeh Seyednasrollah [1], Emad Arbash[1], Satu Hänninen [2], Riikka Kivelä [3,4], Olli Carpen[2,5] & Pipsa Saharinen [1,3,6] ✉

Endothelial cells (ECs) form a tissue-specific barrier for disseminating cancer cells in distant organs. However, the molecular regulation of the ECs in the metastatic niche remains unclear. Here, we analyze using scRNA-Seq, the transcriptional reprogramming of lung ECs six hours after the arrival of melanoma cells in mouse lungs. We discover a reactive capillary EC cluster (rCap) that increases from general capillary ECs in response to infiltrating cancer cells. rCap is enriched for angiogenic and inflammatory pathways and is also found in human lung datasets. The JAK-STAT activated oncogenic *Pim3* kinase is a marker of rCap, being upregulated in spontaneous metastasis models. Notably, PIM inhibition increases vascular leakage and metastatic colonization and impairs the EC barrier by decreasing the junctional cadherin-5 and catenins α, β and δ. These results highlight the pulmonary endothelium's plasticity and its protection by PIM3, which may impair the efficacy of PIM inhibitors in cancer therapies.

Distant metastases present a significant hurdle for the efficacy of anti-cancer therapies. In distant metastasis, invasive cancer cells leave the primary tumor and travel via the blood circulation to colonize distant tissues[1]. Circulating tumor cells (CTCs) become entrapped within organ-specific capillaries before crossing the capillary wall into the tissue, initiating genetic programs of metastatic growth or remaining dormant[2,3]. Metastatic progression is shaped by interactions between tumor cells and their microenvironment, which consists of heterotypic cells including endothelial cells (ECs) of the vascular metastatic niche[4,5]. ECs forming the inner lining of blood capillaries act as a physical barrier creating an obstacle for tumor cell extravasation[2]. Additionally, endothelium is known to express angiocrine factors that affect tumor progression in the metastatic niche[6]. These specialized niches are critical for metastasis; however, their regulation by ECs and other cell types is incompletely understood[2].

The first capillary bed where CTCs become entrapped depends in part on the anatomical patterns of blood circulation. Venous circulation from most organs drains into the lungs, while veins from the gut drain into the liver, promoting metastasis in these organs[7]. Additionally, tissue-specific features such as the low permeability of continuous pulmonary capillary EC junctions and the greater permeability of the liver sinusoids, influence organ colonization[8]. Detailed maps of organ-specific capillaries have been recently generated using single-cell mRNA sequencing (scRNASeq). In the lungs, alveolar capillaries have been shown to consist of two types of endothelial cells: general capillary ECs (gCap), specialized for stem/progenitor cell function, and aerocytes (aCap), specialized for gas exchange[9]. In disease states, most transcriptional changes have been found to occur within capillary beds and in consistent, the alveolar specialization is diminished in lung adenocarcinoma[9,10].

Proviral Integration site for Moloney murine leukemia virus (PIM) kinases are oncogenic serine/threonine kinases overexpressed in many cancers[11]. PIM kinases promote cancer cell survival by inhibiting apoptosis for example through phosphorylation of substrates like

[1]Translational Cancer Medicine, Research Programs Unit, Biomedicum Helsinki, University of Helsinki, Helsinki, Finland. [2]Systems Oncology, Research Programs Unit University of Helsinki, FinlandHelsinki. [3]Wihuri Research Institute, Biomedicum Helsinki, Helsinki, Finland. [4]Faculty of Sport and Health Sciences University of Jyväskylä, Jyväskylä, Finland. [5]Pathology/HUS Diagnostic Center, Helsinki University Hospital, Helsinki, Finland. [6]Department of Biochemistry and Developmental Biology, Faculty of Medicine University of Helsinki, Helsinki, Finland. [7]These authors contributed equally: Niina M. Santio, Keerthana Ganesh. ✉e-mail: pipsa.saharinen@helsinki.fi

BAD, and by enhancing metabolism and motility via NOTCH1[12–14]. Lacking regulatory domains, PIM kinases are primarily controlled at the transcriptional level, resulting in their constitutive activation when expressed. Although initially promising as therapeutic targets, PIM inhibitors have not met expectations in clinical trials[15], and the role of endothelial PIM kinases remains relatively unexplored.

Here, we characterize the vascular metastatic niche of the lungs during the first steps of distant metastasis formation. Using scRNASeq we analyze the pulmonary EC transcriptome six hours after metastatic melanoma injection into the mouse circulation. We identify a reactive capillary EC cluster (rCap) that is enriched in the metastatic mouse lungs and is also found in human lungs. *Pim3* is identified as a marker of the rCap, being also upregulated during spontaneous metastasis in mice. Unexpectedly, we found that PIM inhibition increased vascular leakage and metastatic colonization in mice, via dismantling of the key adherens junction protein Cadherin5 (CDH5) - catenin complex[16]. These results highlight the tumor-induced plasticity of the alveolar capillary endothelium and reveal PIM3 as a regulator of EC barrier integrity, which may explain the lack of efficacy of PIM inhibition so far in clinical cancer trials[17,18].

## Results

### Reactive capillary ECs are enriched in metastatic lungs

To investigate the regulation of the EC barrier during metastatic colonization of the lungs, we used the highly invasive murine melanoma model B16-F10 to induce metastases in syngeneic C57BL/6 mice[19]. Due to the rarity of CTCs arrested in tissues after spontaneous metastasis, we injected CellTracker green-labeled melanoma cells intravenously (i.v.) in mice via the tail vein and followed lung colonization. A few minutes after their injection, melanoma cells were detected in the circulation and within lung capillaries as cell clusters (Supplementary Fig. 1a–c). By 6 h ~70% of the melanoma cells were cleared from the lungs, and those remaining were present as smaller clusters and as single cells while the numbers further dropped by 16 h (Supplementary Fig. 1b). Few metastatic cells remaining in the lungs formed metastatic nodules starting from day 3 and increasing until the endpoint on day 11 (Supplementary Fig. 1b–d). Liver metastasis occurred significantly slower, and by day 9 only a few small metastatic nodules appeared on the surface of the liver (Supplementary Fig. 1e), as expected following tail vein injection of B16-F10 in contrast to induction of liver metastasis by portal vein injection[20].

To investigate the changes in the EC transcriptome during the first steps of metastatic colonization, PECAM1+ (also known as CD31) PTPRC- (also known as CD45) ECs and labeled melanoma cells were isolated using flow cytometry from the lungs at 6 and 30 h after melanoma cell (or PBS as a control) injection and subjected to scRNASeq (Fig. 1a, b, Supplementary Fig. 2a–e). The EC and melanoma cell subsets were retrieved from the quality-filtered, normalized and integrated dataset containing control and metastatic samples and subsequently reclustered for analysis (Fig. 1c, Supplementary Fig. 2f–j). Altogether, 21220 ECs were obtained in three independent experiments from control and 19,266 ECs from metastatic lungs at 6 h, while 6889 ECs were obtained in a single experiment at 30 h after melanoma injection (Fig. 1c). 91 B16-F10 cells were obtained, reflecting the low number of melanoma cells in the lungs 6 h post injection (Fig. 1c, Supplementary Fig. 1b). Inflammatory cells, including Cd68+ macrophages and S100a8+ neutrophils were similarly present in the lungs at baseline and 6 h post melanoma injection, and were excluded from the analysis (Supplementary Fig. 3a–d).

Previously reported markers were used to annotate the lung EC clusters as general (gCap) and aerocyte capillaries (aCap), arteries (artery), veins (vein), lymphatic (LEC) as well as interferon rich ECs (hIFN), previously found in lung tumors and different mouse organs (Fig. 1c–e)[21–23]. Interestingly, an additional EC cluster was identified, which expressed unique cluster markers in addition to gCap markers

and was increased in cell number 6 h after melanoma injection in mice, hence, we named it the reactive capillary EC (rCap) (Fig. 1c–f, Supplementary Data 1). To identify the origin of the rCap cluster, we used RNA velocity to predict the directionality of the gene expression changes by quantifying the relationship between the abundance of unspliced and mature mRNA transcripts (Fig. 1g). The velocity analysis revealed that rCap cluster was derived from the gCap, in consistent with the expression of gCap markers by rCap. rCap cluster markers were upregulated at 6 h post injection of melanoma, however, the markers were downregulated again by 30 h (Fig. 1h, i). The temporal regulation of rCap marker gene expression is in-line with the decrease of melanoma cells in the lungs 30 h post injection (Supplementary Fig. 1b). We confirmed significant upregulation of top rCap markers, including *Pim3, Bcl3, Inhhb, Osmr, Tmem252, Adamts9* and *Scarb1*, in isolated mouse lung ECs 6 h after B16-F10 injection using RT-qPCR (Fig. 1j). These results indicate that circulating melanoma cells induced rapid transcriptional changes in gCap, leading to an expansion of a reactive capillary EC cluster already 6 h after tumor cell arrival in the lungs.

### rCaps upregulate angiogenic and inflammatory pathways

To visualize paracrine signaling networks of ECs in the metastatic lungs, the CellChat tool of inference and analysis of cell-cell communication was used[24]. Interestingly, the rCap cluster was enriched as the source of signaling interactions in metastatic lungs when compared to the other EC clusters (Fig. 2a). Further analysis identified the upregulation of VEGF, NOTCH, collagen, endothelial cell adhesion protein ESAM and activin pathways in rCap (Fig. 2b–d, Supplementary Data 2), while melanoma cells were found to communicate with all ECs (Supplementary Fig. 4). Notably, whereas rCap became a sender of *Vegfa* in the metastatic lungs, melanoma cells expressed *Vegfb* (Fig. 2b–d, Supplementary Fig. 4b, c), suggesting potential competition in receptor binding on the recipient cells[25]. The rCap also induced the NOTCH pathway ligand and tip cell marker *Dll4*, which signals through NOTCH1 during sprouting angiogenesis[26] as well as collagen 4a2 (*Col4a2*), expressed by angiogenic breach ECs[23] (Fig. 2b–d). Moreover, rCap expressed *Inhbb*, which has been previously identified in lung metastases[27,28] as well as the tight junction protein *Esam* implicated in EC barrier function[29,30] (Fig. 2b, d). Melanoma cell expressed *Mif* signaled to rCap and venous ECs through *Ackr3* (a.k.a CXCR7) receptor (Supplementary Fig. 4b, c), which has been shown to mediate inflammatory cell interactions with ECs via adhesion molecule expression[31–33]. These results indicate that tumor cells rewire paracrine EC signaling pathways, enriched in rCap in the metastatic lungs.

Kyoto Encyclopedia of Genes and Genomes (KEGG) and Hallmark gene set enrichment analysis (GSEA) of the differentially expressed (DE) rCap genes between the metastatic and control lungs revealed activation of inflammatory, cytoskeleton, adhesion and tight junction pathways (Fig. 2e, f, Supplementary Data 3–5, Supplementary Table 1). Furthermore, rCap upregulated angiogenic markers, such as the glycolytic enzyme *Pfkfb3*, previously found in regulation of the EC barrier and metastasis[34] as well as *Jak2*. Moreover, GSEA revealed activation of the JAK-STAT signaling pathway in rCap (Fig. 2f, g) although it was enriched also in other EC clusters (Supplementary Data 3–5, Supplementary Table 1). The JAK-STAT pathway is known to regulate metastasis and tumor angiogenesis[35–37], and interestingly, two prominent rCap markers *Pim3* and *Bcl3* are direct transcriptional targets of the JAK-STAT pathway[38,39] (Fig. 2h). In conclusion, these results indicate that the rCaps were increased in response to infiltrating melanoma cells, being characterized by the transcriptional upregulation of angiogenic markers and the JAK-STAT-PIM3-BCL3 pathway.

### rCap markers localize in the vicinity of tumor cells

To locate the rCaps within the lungs, we used RNAscope® in situ hybridization (ISH) of the rCap markers *Pim3, Bcl3* and *Inhbb*

(Fig. 3a–c). An antibody against melanoma marker PMEL was used to detect the B16-F10 cells and *Pecam1* probe to identify ECs. Interestingly, *Inhbb* and *Bcl3* showed low expression in the control lungs, while *Pim3, Inhbb* and *Bcl3* were enriched in the metastatic lungs, especially in the vicinity of melanoma cells when compared to regions distant to melanoma cells (Fig. 3d–h, Supplementary Fig. 5). 46% and 16% of *Pecam1*+ ECs were positive for both *Pim3* and *Bcl3* or *Pim3* and *Inhbb*, respectively, indicating tumor-induced upregulation of the rCaps 6 h

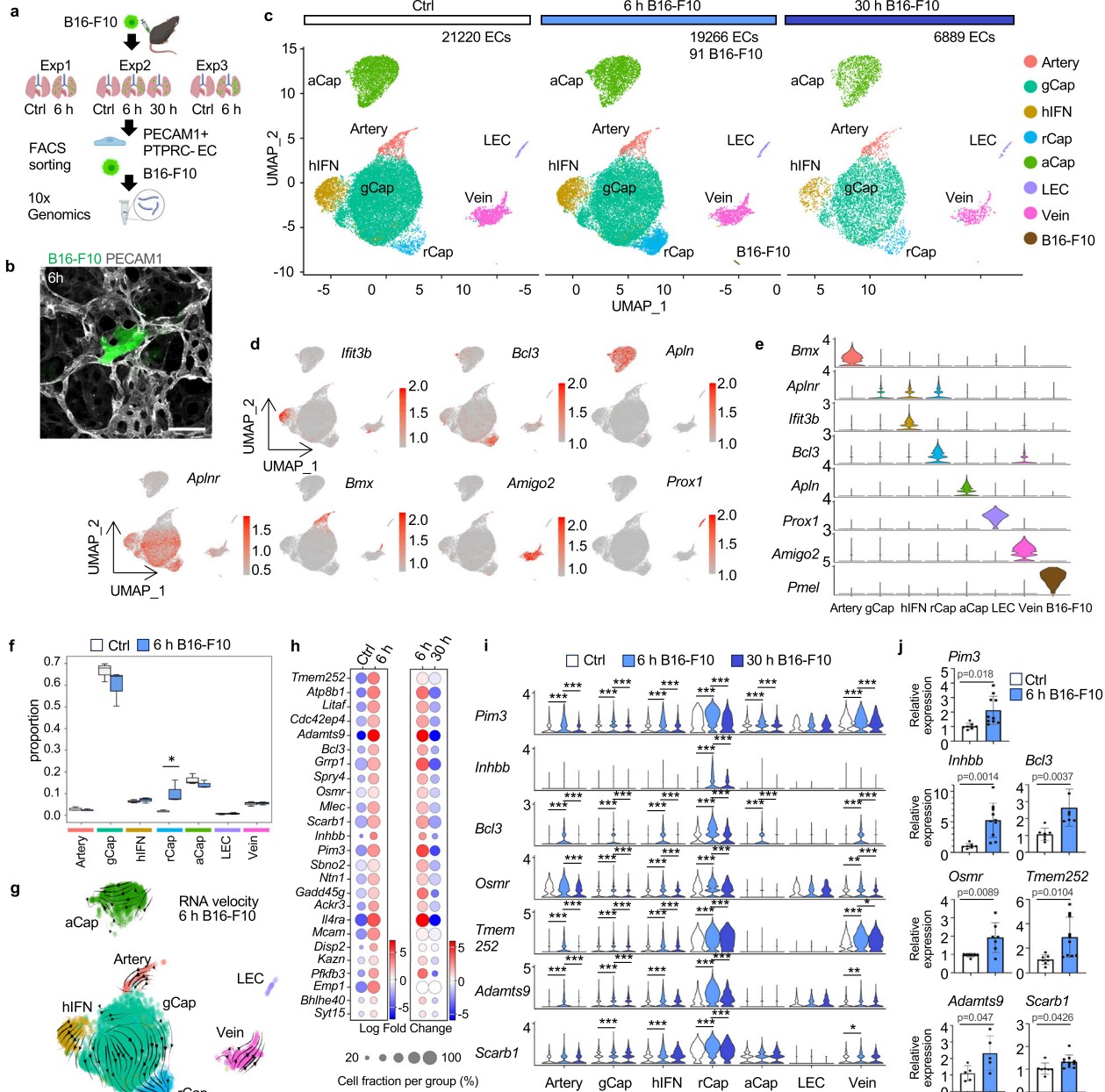

**Fig. 1 | Single-cell RNA-Seq reveals reactive capillary endothelial cells (rCap) in the lungs 6 h after cancer cell injection. a** B16-F10 melanoma cells or PBS (Ctrl) were intravenously (i.v.) injected into C57BL/6 mice, and endothelial (ECs) and B16-F10 cells were isolated from the lungs after 6 h or after 30 h and subjected for scRNA-Seq. *n* = 3 independent experiments at 6 h, *n* = 1 exp at 30 h. Cells from two or three mice were pooled per sample in experiments 1 and 2–3, respectively. **b** Representative image of a 150 μm thick lung section showing a CellTracker green labeled B16-F10 cell in the lung capillary stained for PECAM1 6 h post-injection. *n* = 3 mice injected with B16-F10, compared to 3 Ctrl mice. **c** Uniform Manifold Approximation and Projection (UMAP) of clustered scRNA-Seq data of the lung EC and melanoma subsets. **d, e** EC cluster markers in the integrated data containing controls and all time points. **f** Analysis of EC composition per cluster between Ctrl and 6 h post-injection of B16-F10 using scCODA (false discovery rate, FDR > 0.05*). *n* = 3 independent experiments. The median is indicated by a line, bounds of the

box represent the first and third quartiles, and the whiskers the smallest and largest data points within 1.5 interquartile ranges from the bounds. **g** RNA velocity analysis at 6 h time point. **h** Fold change of the top 25 rCap cluster markers, ordered based on adjusted *p* value. **i** rCap marker expression in EC clusters in control lungs, and at 6 h and 30 h after B16-F10 injection. *n* = 3 independent experiments (6 h), *n* = 1 (30 h) (g-i). **j** RT-qPCR of rCap markers in ECs isolated from control lungs (*n* = 7 mice for all markers) and 6 h after B16-F10 i.v. injection (*n* = 10 for *Pim3, Inhbb, Tmem252* and *Scarb1* each, *n* = 6 for *Bcl3*, *n* = 7 for *Osmr*, *n* = 5 for *Adamts9*). Each dot represents the average from one mouse, values were normalized to *Hprt*. General capillary, gCap; high interferon EC, hIFN; aerocyte capillary, aCap; lymphatic EC, LEC. Wilcoxon rank-sum test (**h, i**) (two-sided in **i**), two-tailed unpaired *t*-test (**j**), *p* < 0.05*, <0.01**, <0.001***. Data are presented as mean values +/- SD. Scale bar, 10 μm (b). Illustration created in Biorender.com (**a**). Source data are provided as a Source Data file.

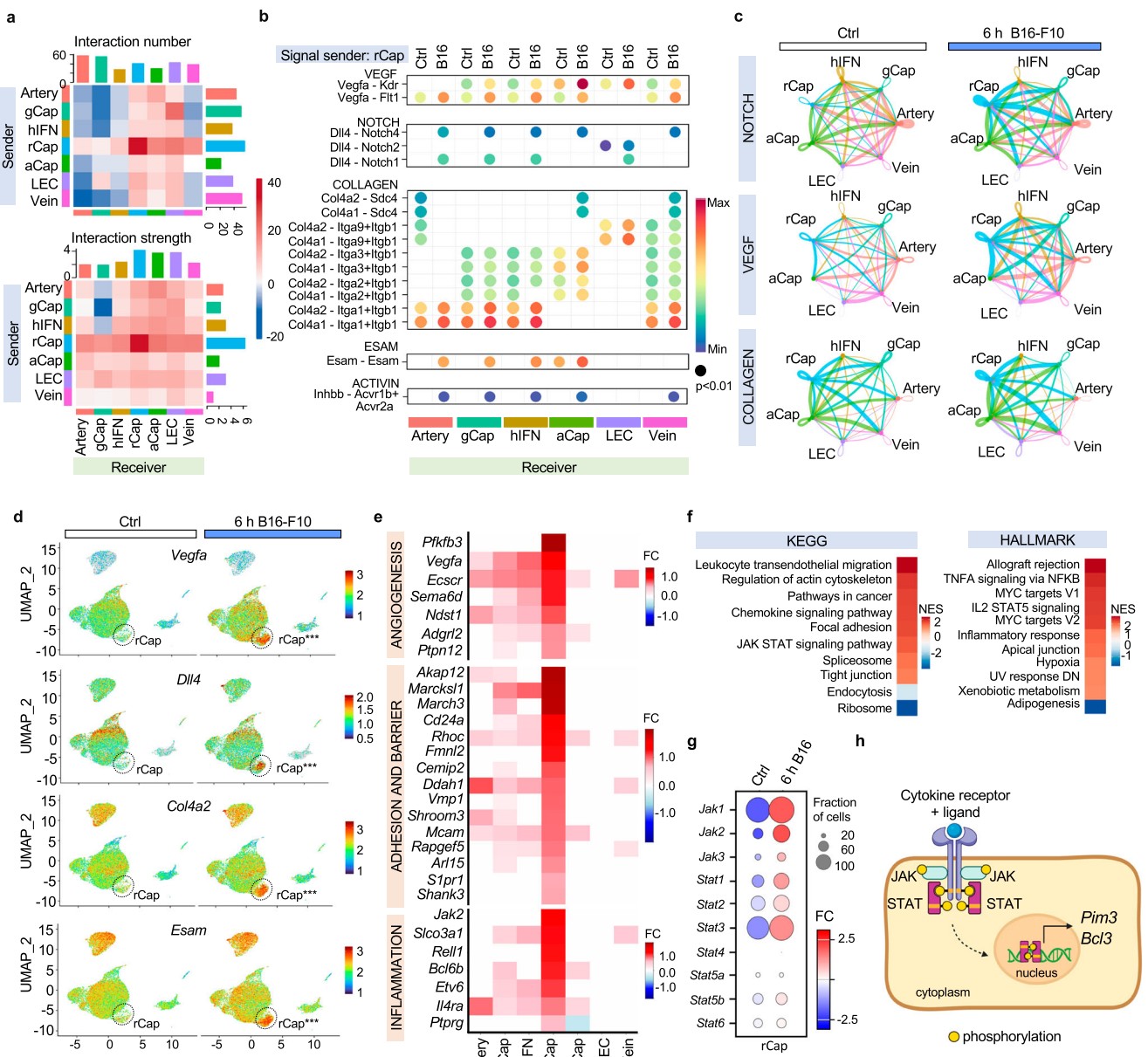

**Fig. 2 | Cell-cell communication and gene set enrichment analysis reveal angiogenic and inflammatory signatures in rCap.** **a** Overview of signaling interactions between lung endothelial cells (ECs) using CellChat, 6 h after B16-F10 intravenous (i.v.) injection in mice compared to PBS (Ctrl). $n = 3$ independent experiments. **b** CellChat communication probability of upregulated ligand-receptor interactions originating from reactive capillary EC (rCap) in the metastatic lungs compared to Ctrl, $p < 0.01$ in all. **c** Signal strength of indicated pathways between EC clusters in Ctrl and 6 h samples using circle plots. **d** Feature plots showing expression of selected rCap enriched ligands 6 h after B16-F10 i.v. injection in mice compared to Ctrl. rCap cluster is highlighted. Significant changes marked***

(two-sided Wilcoxon rank-sum test for fold change between 6 h and Ctrl, adjusted $p$-value < 0.05) **e** Differentially expressed (DE) rCap genes 6 h after B16-F10 i.v. injection in mice compared to Ctrl, selected among the top 100 most significant based on adjusted $p$-value. FC was set to 0, when adjusted $p$-value > 0.001. **f** Kyoto Encyclopedia of Genes and Genomes (KEGG) and Hallmark gene set enrichment analysis of the rCap DE genes between 6 h metastatic and control lung. **g** Gene expression changes of JAK-STAT pathway genes in rCap at 6 h. **h** Schematic representation of the JAK-STAT-PIM-BCL3 pathway created in Biorender.com. Source data are provided as a Source Data file. aCap Aerocyte capillary, gCap general capillary, hIFN high interferon EC, LEC lymphatic EC.

after melanoma injection (Fig. 3i–l). Moreover, PIM3 protein levels were increased in the lung lysates 7 h after i.v. injection of B16-F10 when compared to controls (Fig. 3m, n). To sum up, rCap markers localize in the lung endothelium in the vicinity of infiltrated cancer cells.

**rCap markers are upregulated during spontaneous metastasis**

To validate rCap regulation during metastasis, we used spontaneous metastasis models, which better recapitulate the metastatic process compared to i.v. injection of tumor cells. To this end, we inoculated

B16-F10 subcutaneously into C57BL/6, which resulted in the formation of vascularized tumors by day 14 (Fig. 4a, Supplementary Fig. 6a–c). CTCs were detected in blood samples 14 d, but not 7 d, after tumor cell inoculation (Fig. 4b). In concordance, *Pim3*, *Bcl3* and *Inhbb* were upregulated in lung ECs 14 d, but not 7d after cancer cell inoculation (Fig. 4c). However, no lung metastases were detected up to 14 d in the lungs, suggesting rCap marker upregulation by CTCs (Supplementary Fig. 6d).

Next, we used the orthotopic 4T1 mammary tumor model, which forms vascularized tumors and metastases to both the lungs and the

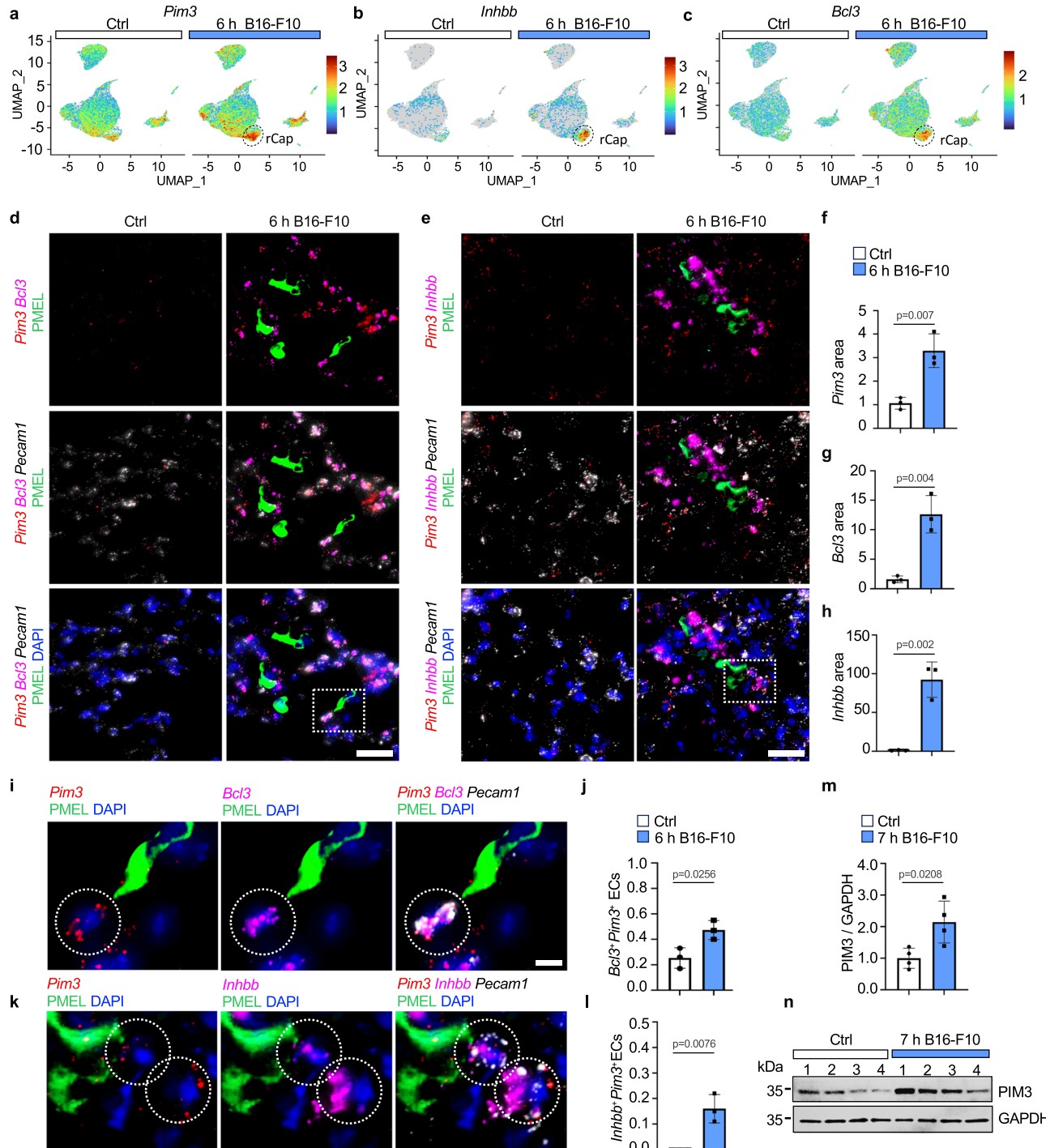

**Fig. 3 | rCap markers are enriched in ECs in metastatic lungs.** Visualization of reactive capillary endothelial cell (rCap) markers *Pim3* (**a**), *Inhbb* (**b**) and *Bcl3* (**c**) expression in the integrated scRNASeq data from mouse lungs, 6 h post intravenous (i.v.) injection of B16-F10 melanoma cells or PBS (Ctrl). *n* = 3 independent experiments. **d, e** Representative images of RNAscope™ in situ hybridization (ISH) and immunohistochemistry of lung sections from similarly injected mice. *Pim3* (**d, e**), *Bcl3* (**d**), *Inhbb* (**e**), and *Pecam1* (**d, e**) were detected using ISH, melanoma cells using anti-PMEL antibody and nuclei by DAPI. *n* = 3 mice per group. Quantification of *Pim3* (**f**), *Bcl3* (**g**) and *Inhbb* (**h**) signal area in the lungs, normalized to nuclear area, relative to Ctrl. *n* = 3 mice per group. **i–l** Representative higher magnification images from samples in (**d, e**), showing coexpression of *Pim3* with *Bcl3* (**i**) or *Inhbb* (**k**) transcripts near cancer cells. Quantification of *Pim3*+*Bcl3*+ (**j**) and *Pim3*+*Inhbb*+ (**l**) double positive *Pecam1*+ cells normalized to all *Pecam1*+ ECs from Ctrl and 6 h B16-F10 lung. *n* = 3 mice per group. **m, n** Quantification of PIM3 protein levels in ECs isolated from Ctrl and 7 h B16-F10 i.v. injected mouse lungs, normalized to GAPDH. *n* = 4 mice per group. Wilcoxon rank-sum test (**a–c**), two-tailed unpaired *t*-test (**f–h, j, l, m**). Data are presented as mean values +/- SD. Scale bars 25 µm (**d, e**) and 5 µm (**i, k**). Source data are provided as a Source Data file.

liver in BALB/c mice (Fig. 4d–h, Supplementary Fig. 6e–g). The formation of lung metastasis was slower compared to the liver: whereas small metastatic nodules were detected in the liver already after 6 days, visible lung metastases appeared 3–4 weeks after tumor cell implantation (Fig. 4e, g). Interestingly, *Pim3, Bcl3* and *Inhbb* were upregulated in ECs isolated from the lungs and liver 14 d after tumor cell inoculation, a time point where CTCs were present, as indicated by the liver metastases (Fig. 4f, h). Notably, B16-F10

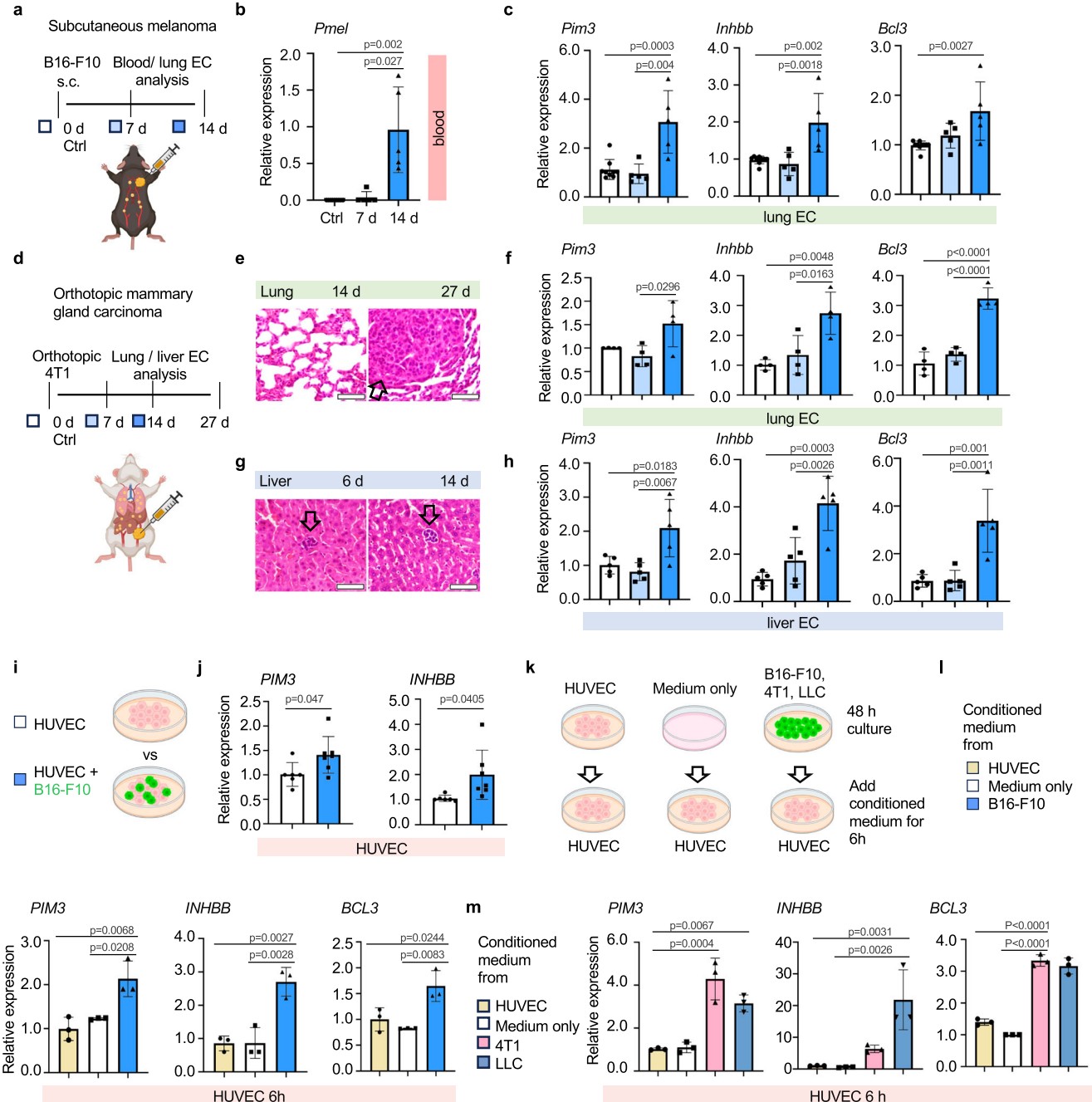

**Fig. 4 | rCap markers are upregulated in spontaneous metastasis models and by tumor cell secreted factors. a–c** B16-F10 melanoma cells or PBS (Ctrl) were injected subcutaneously (s.c.) into C57BL/6 mice (**a**). RT-qPCR for melanoma marker *Pmel* in blood samples (**b**) and for *Pim3, Inhbb* and *Bcl3* in isolated lung ECs from Ctrl and tumor bearing mice at 7 and 14 days after tumor implantation (**c**). *n* = 5 mice per group. **d–h** 4T1 mammary carcinoma cells were implanted orthotopically into BALB/c mice and analyzed at indicated time points (**d**). H&E staining of lung (27 d after tumor initiation) (**e**) and liver metastasis (6 d and 14 d after tumor intiation) (**g**) (arrows indicate metastatic nodules). RT-qPCR for *Pim3, Inhbb* and *Bcl3* in isolated lung (*n* = 4 mice per group) (**f**) and liver ECs (*n* = 5 mice per group) (**h**). **i, j** Mono- and cocultures of human umbilical vein endothelial cells (HUVEC) with B16-F10 melanoma cells (**i**). RT-qPCR of *PIM3* and *INHBB* in sorted HUVECs. *n* = 6 independent experiments for monocultures, *n* = 7 for B16-F10-HUVEC cocultures (**j**). HUVECs were treated for 6 h with conditioned medium incubated for 48 h on cancer cell or HUVEC cultures or empty plates, as indicated (**k**), and analyzed using RT-qPCR. *n* = 3 independent experiments (**l, m**). RT-qPCR results are shown relative to Ctrl, normalized to *Hprt*. One-way Anova with multiple comparisons (**b, c, f, h, l, m**), two-tailed unpaired *t*-test (**j**). Data are presented as mean values +/- SD. Scale bars 50 µm (**g, e**). Illustrations created in Biorender.com (**a, d, i, k**). Source data are provided as a Source Data file.

melanoma cells are not arrested in the liver sinusoids 6 h after tail vein injection, and consistently, rCap markers were not upregulated in the liver at this time point (Supplementary Fig. 6h). These results suggest that during spontaneous metastasis, tumor cells in the secondary organs or in circulation induce the upregulation of rCap markers, whereas this effect was not induced from distance by the primary tumor.

## Tumor cell-secreted factors induce rCap markers

To further investigate the mechanism by which tumor cells induce upregulation of rCap markers, human umbilical vein ECs (HUVECs) were cocultured with B16-F10 melanoma cells stably expressing green fluorescent protein (GFP), followed by sorting of non-labeled ECs and GFP-positive melanoma cells using flow cytometry (Fig. 4i–j, Supplementary Fig. 6i). Interestingly, the expression of *PIM3* and *INHBB* was

increased in sorted HUVECs 16 h after coculturing with tumor cells, as compared to HUVEC monocultures, indicating that tumor cells were sufficient to induce upregulation of rCap markers (Fig. 4j). Moreover, conditioned media from B16-F10, 4T1 or Lewis lung carcinoma (LLC) cultures similarly induced *PIM3, INHBB* and *BCL3* expression, indicating that secreted factors of multiple cancer cell types induced rCap marker expression in cultured ECs (Fig. 4k−m, Supplementary Fig. 6j). Therefore, we compared HUVECs cultured in either control or B16-F10 conditioned media for 16 h using bulk RNA-Seq (Supplementary Fig. 7). GSEA of DE genes revealed upregulated KEGG and Hallmark pathways in response to melanoma secretome, including hypoxia, glycolysis, angiogenesis, and the JAK-STAT pathway (Supplementary Fig. 7). Thus, results from cultured ECs support the conclusion that melanoma cell secreted factors shape the lung capillary endothelium, leading to a specific transcriptional signature and among others, activation of the JAK-STAT pathway.

## rCaps show similarity to mouse tumor vasculature

The increased number of rCap in metastatic vs healthy lungs and the upregulation of angiogenic markers in rCap prompted us to explore whether rCaps showed similarity to tumor vascular endothelium. Therefore, we first obtained three LLC tumor EC (TEC) scRNASeq datasets and a single normal EC (NEC) dataset from mouse Lung Tumor EC Tax on-line source[23] (Supplementary Fig. 8a). The EC clusters of integrated TEC and NEC datasets were annotated based on markers by Goveia et al. (Supplementary Fig. 8b−e)[23]. Projection of the top 50 rCap markers to the integrated TEC and NEC datasets showed enrichment of rCap markers in the TECs, especially in the tumor post capillary venule (PCV), tumor capillary (tCap) and breach EC clusters (Supplementary Fig. 8f). *Pim3* was among the top 50 tCap markers, while the rCap marker *Adamts9* was a marker of PCVs (Supplementary Fig. 8g, h). Similarly, projection of the PCV and breach TEC markers showed enrichment in rCap in our dataset, 6 h post melanoma injection, whereas tCap markers were expressed by several capillary clusters (Supplementary Fig. 8i). These results indicate that although rCap markers are widely expressed in the tumor capillaries, rCaps show highest similarity to PCV and breach ECs.

## rCaps form a distinct cluster in human lungs

To investigate whether rCap can be found also in the human lungs, we used cross-species comparison of scRNASeq data from our mouse lung EC dataset and ECs from the Human Lung Cell Atlas (HLCA) (Supplementary Fig. 9a)[40,41]. Since rCap was present also in the healthy mouse lungs, albeit increased after melanoma injection, we utilized scRNASeq data of the EC subset from healthy lung and normal lung tissue adjacent to lung tumor from carcinoma patients[40,41]. The integrated human ECs were clustered and annotated based on previously reported markers as aCap, gCap, artery, pulmonary (veinP) and systemic vein (veinS), LEC, and the intermediate capillary identities of both aCap (Int-aCap) and gCap (Int-gCap) (Supplementary Fig. 9b, c)[9]. Notably, our analysis identified an additional gCap cluster, which expressed specific markers including *ICAM1, EMP1, ARID5A* and *PIM3* and which showed an enrichment of the rCap gene signature consisting of the top 50 mouse rCap markers, suggesting it represents the human rCap (Supplementary Fig. 9b−e). Visualization of the top human rCap markers (*ICAM1, EMP1* and *ARID5A)* and one of the top mouse markers *PIM3* showed strong expression in the human rCap cluster, along with some expression in veins and Int-aCap (Supplementary Fig. 9e).

To further confirm the presence of the rCap cluster in human lungs, we integrated the human lung EC data with our control and 6 h B16-F10 i.v. injected metastatic mouse lung EC data (Supplementary Fig. 9f). Using RPCA integration, we found successful cross-species mixing, while the previously annotated lung EC clusters were retained, except for the hIFN mouse ECs, which were not identified in human

and the human systemic vein ECs which were not detected in mouse lung ECs (Supplementary Fig. 9g). However, after data integration, the intermediate Int-aCap and Int-gCap clusters were identified also in mouse lungs. Importantly, the rCap cluster was identified in both mouse and human data, sharing ~50% of the rCap cluster markers (Supplementary Fig. 9g, h). Visualization of the major human rCap markers *EMP1, ICAM1, ARID5A* and *PIM3* showed strong expression in the rCap and Int-aCap also after integration with mouse data (Supplementary Fig. 9i, j). These results indicate that rCap form a distinct capillary EC cluster both in mouse and human lungs, with shared marker genes.

## PIM3 regulates the endothelial barrier

The endothelial JAK-STAT pathway is known to regulate cancer metastasis[35–37], but its downstream targets, the PIM kinases, have not been studied in detail in ECs. In contrast, PIM kinases are well-known oncogenes conferring cancer cell survival, and therefore, have been investigated as anti-cancer drug targets in clinical cancer trials. However, the efficacy of PIM inhibitors in cancer control has been lower than anticipated, leading to the discontinuation of several inhibitors in clinical development, and prompting a search for more selective inhibitors and optimized combination therapies[17,18]. Our results revealed the tumor-induced endothelial JAK-STAT-PIM-BCL3 axis both in vitro and in vivo, with upregulation of also *Pim1* in rCap, albeit less than *Pim3*, whereas *Pim2* was not expressed in the lung ECs (Supplementary Fig. 10a, b). In line with results from lung ECs, *PIM3* was the most prominently expressed PIM family member in HUVEC and in dermal blood microvascular ECs (BECs), whereas *PIM1* and *PIM2* were expressed at low level (Supplementary Fig. 10c).

To understand the function of PIM3, we inhibited PIM kinases in HUVECs and BECs using *PIM3* shRNA and a selective PIM inhibitor, AZD-1208[15]. *PIM3* silencing decreased the adherens junction protein CDH5 on the EC surface and induced gaps in EC-EC junctions as compared to the scrambled control (shScr), whereas total CDH5 levels were comparable to shScr, indicating that PIM3 specifically maintained junctional, cell surface localized CDH5 (Fig. 5a−e, Supplementary Fig. 10d). Furthermore, the tension-sensitive α-catenin (CTNNA1), β-catenin (CTNNB1) and δ-catenin (CTNND1) that connect CDH5 to the actin cytoskeleton were significantly decreased in the cell-cell junctions of shPIM3 ECs when compared to shScr ECs (Fig. 5f−m).

Using electrical cell impedance sensing assay, PIM inhibition using AZD-1208 decreased the barrier function of EC monolayers (Fig. 6a, b). However, no changes in HUVEC confluency or apoptosis were detected in reduced serum containing media during 72 h of PIM inhibition, suggesting that the PIM inhibition-induced decrease in EC barrier function was not caused by impaired EC survival under these conditions (Supplementary Fig. 10e). However, CDH5 was decreased in the cell-cell junctions of AZD-1208 -treated EC monolayers and characterized by gaps in the CDH5 junctions, whereas CDH5 total protein levels were not changed (Fig. 6c−g), in line with results using shPIM3 silencing. These results indicate that PIM3 maintains the integrity of EC-EC junctions in culture.

Analysis of B16-F10 metastatic mouse lungs by sorting PECAM1+ ECs, CellTracker+ melanoma and PECAM1-CellTracker- cell populations showed significantly higher levels of *Pim3* in ECs, as compared to the other lung cell populations (Supplementary Fig. 10f). Therefore, to investigate whether PIM regulates adherens junction integrity also in vivo, we administered AZD-1208 or vehicle orally for 5 days in mice and stained thick lung sections for CDH5 (Fig. 6h−l). Whereas CDH5 staining was continuous between EC-EC junctions in the alveolar capillaries in the control mice, gaps in CDH5 junctions were observed in the lungs of AZD-1208 −treated mice (Fig. 6h−l, Supplementary Movie 1, 2), suggesting, in consistent with our in vitro results, that PIM inhibition decreases CDH5 in the EC-EC junctions also in vivo in mouse lungs.

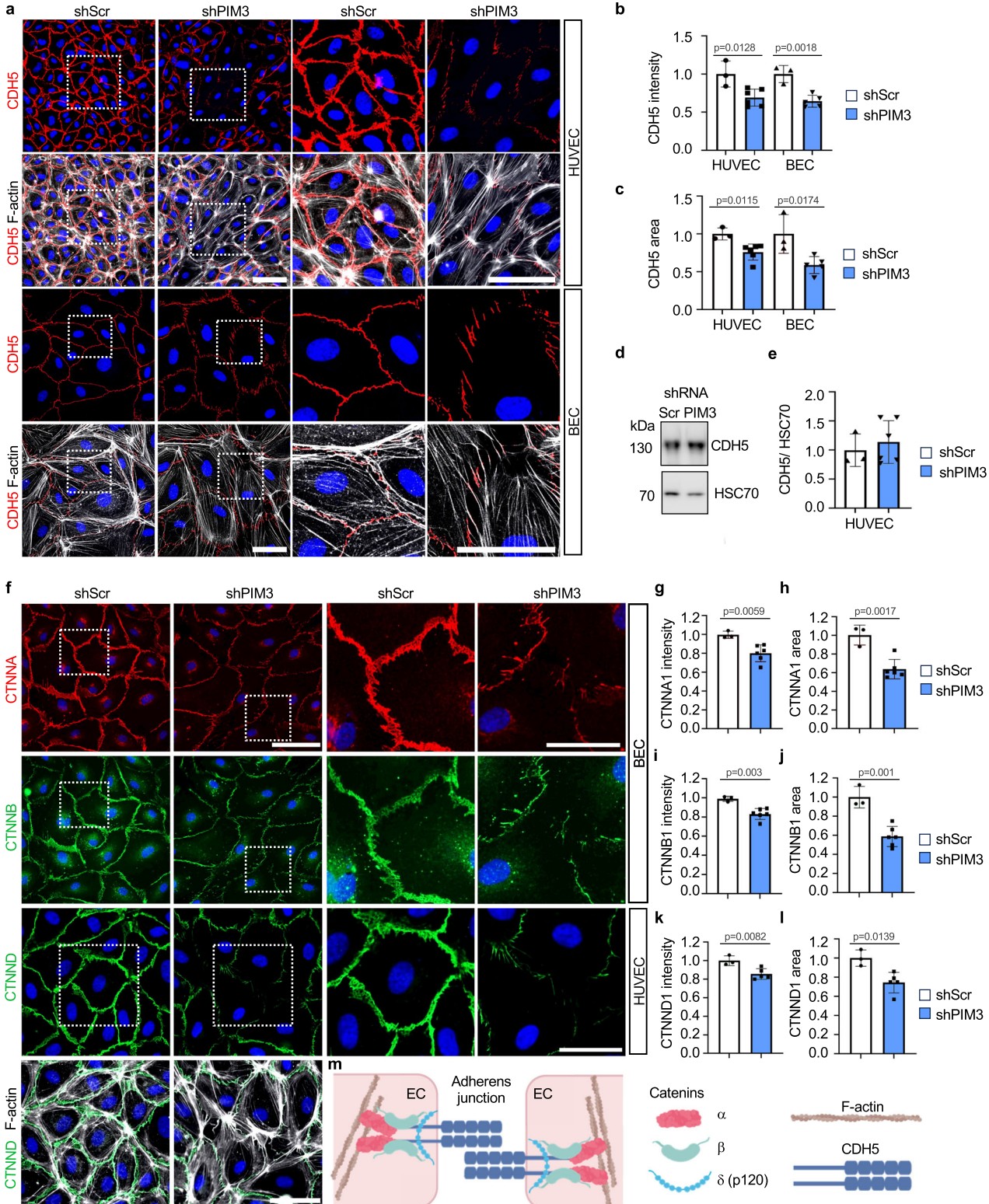

**Fig. 5 | Silencing of *PIM3* dismantles cadherin-5 and catenins from endothelial cell-cell junctions. a** shPIM3 or control (shScr) silenced human umbilical vein endothelial cells (HUVECs) and dermal microvascular endothelial cells (BECs) were stained for vascular endothelial cadherin (CDH5) and F-actin. Nuclei were stained using DAPI. Relative CDH5 signal intensity (**b**) and area (**c**) (normalized to number of nuclei) in HUVEC (*n* = 3 independent experiments for shScr, *n* = 6 independent experiments for shPIM3) and in BEC (*n* = 3 for shScr, *n* = 5 for shPIM3). Western blot (**d**) and quantification (**e**) of CDH5 in HUVECs treated as in (**a**). *n* = 3 independent experiments for shScr, *n* = 6 for shPIM3. **f** shPIM3 or shScr silenced BECs were stained for α- and β-catenin (CTNNA1 and CTNNB1) and HUVECs for δ-catenin

(CTNND1) and F-actin. Nuclei were stained using DAPI. Relative α-catenin (**g, h**), β-catenin (**i, j**) and δ-catenin (**k, l**) signal intensities (per field) and area (normalized to number of nuclei) relative to control (shScr). *n* = 3 for shScr, *n* = 6 for shPIM3 (**g–j**). *n* = 3 for shScr, *n* = 5 for shPIM3 (**k, l**). **m** Schematic representation of CDH5 and α-, β- and δ-catenin in adherens junctions created in Biorender.com. Two-tailed unpaired *t*-test (**b, c, e, g–l**). Data are presented as mean values +/- SD. Data is pooled from independent experiments using two shPIM3 clones in **b, c, e, g–l**. Scale bars 50 μm (**a, f**); 25 μm in close-up images (**a, f**). Source data are provided as a Source Data file.

## PIM inhibition increases vascular leakage and metastasis

To investigate the functional significance of PIM-mediated regulation of EC junction integrity in vivo, we administered AZD-1208 in mice daily for three days and analyzed vascular leakage by injecting 70 kd fluorescent dextran for 10 min i.v. Microscopic analysis showed strong dextran signal in AZD-1208-treated lungs, indicating leaked tracer, whereas control lungs showed low signal (Supplementary Fig. 11a, b), indicating that AZD-1208 impairs the EC barrier function in the lungs.

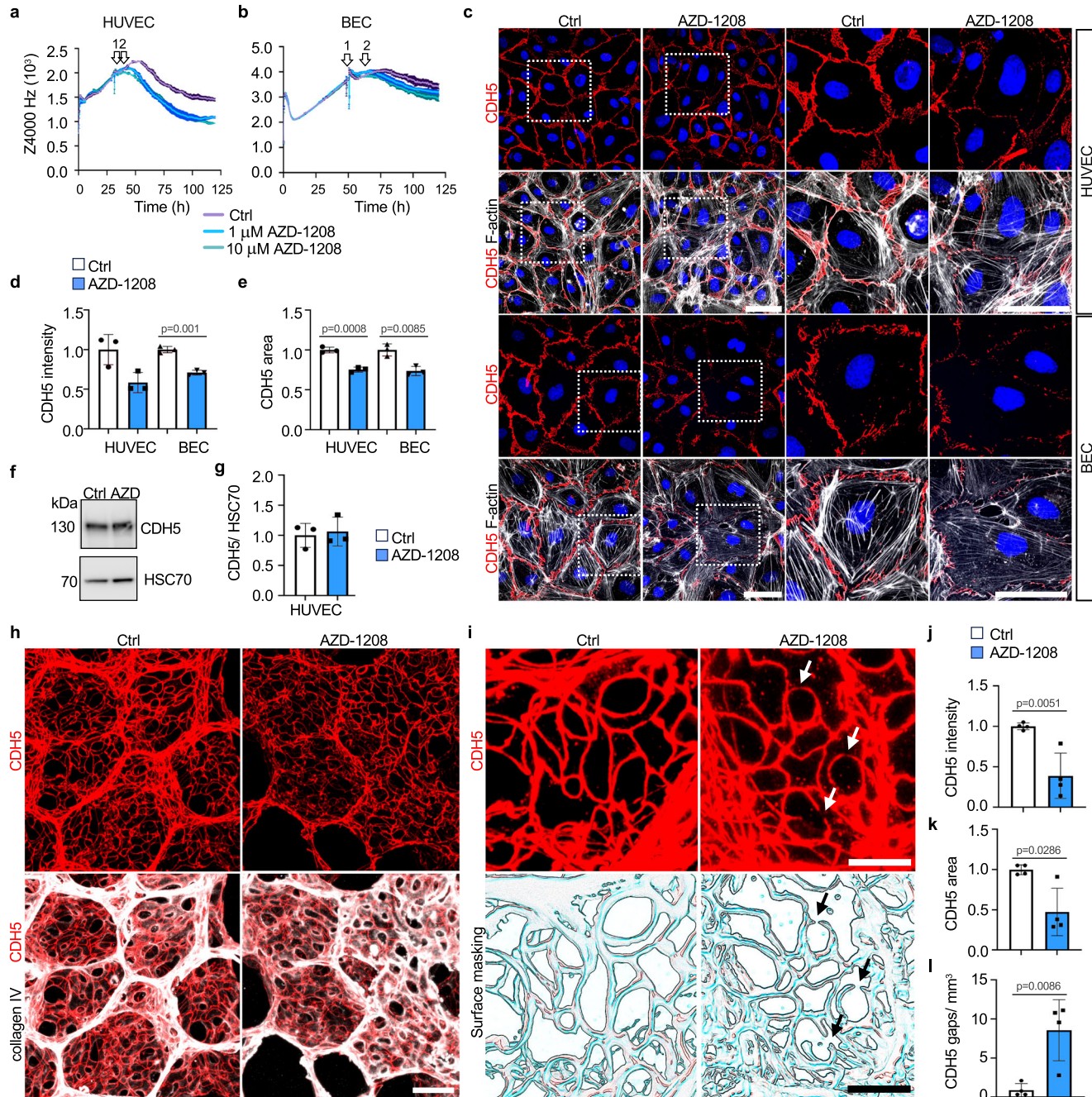

**Fig. 6 | PIM inhibition decreases junctional cadherin-5 and impairs the endothelial cell barrier.** Control or AZD-1208 treated human umbilical vein endothelial cells (HUVEC) (**a**) and human dermal microvascular blood endothelial cells (BEC) (**b**) were analyzed using ECIS electrical cell impedance sensing. Arrows indicate initiation of serum starvation (arrow 1) and initiation of treatment (arrow 2). Shown are representative experiments with triplicate samples with SEM. Significant differences based on three independent experiments ($n = 3$, Ctrl vs AZD-1208 at indicated times after treatment) for HUVEC at 24 h 1 μM ($p = 0.0372$), 10 μM ($p = 0.0026$), at 48 h 1 μM ($p = 0.0456$), 10 μM ($p < 0.0001$) and at 72 h 10 μM ($p = 0.0053$) and for BEC at 48 h 1 μM ($p = 0.0086$), 10 μM ($p = 0.0012$). **c** Representative images of HUVEC and BEC treated with AZD-1208 (1 μM) or 0.1% DMSO (Ctrl) for 24 h in reduced 2.5% serum, and stained for CDH5, F-actin and nuclei (DAPI). Relative CDH5 signal intensity (per field) (**d**) and area (normalized to number of nuclei) (**e**). $n = 3$ independent experiments. **f, g** CDH5 Western blot and quantification of HUVEC treated as in (**c**). $n = 3$ independent experiments. **h** AZD-1208 (30 mg kg⁻¹) or vehicle (Ctrl) was orally administered daily for 5 days. CDH5 and collagen IV (Col IV) were stained in thick lung sections. Shown are maximum intensity projections of confocal z-stacks. **i** Magnification of maximum intensity projection of CDH5 stained lung sections (top) with surface masking (below). Arrows indicate gaps in CDH5 staining. Quantification of CDH5 intensity (**j**) and area (**k**), and number of gaps in CDH5 staining (**l**) as explained in materials and methods. $n = 4$ independent experiments. Mixed-effects analysis (**a**) and two-way ANOVA (**b**) both with multiple comparisons, two-tailed unpaired $t$-test (**d, e, g, j, l**), two-sided Mann-Whitney U test (**k**). Data are presented as mean values ⁺/⁻ SD (**d, e, g, j–l**), or ⁺/⁻ SEM (**a, b**). Scale bars 50 μm (**c**); 25 μm (**h**, close-up images in **c**) and 10 μm (**i**). Source data are provided as a Source Data file.

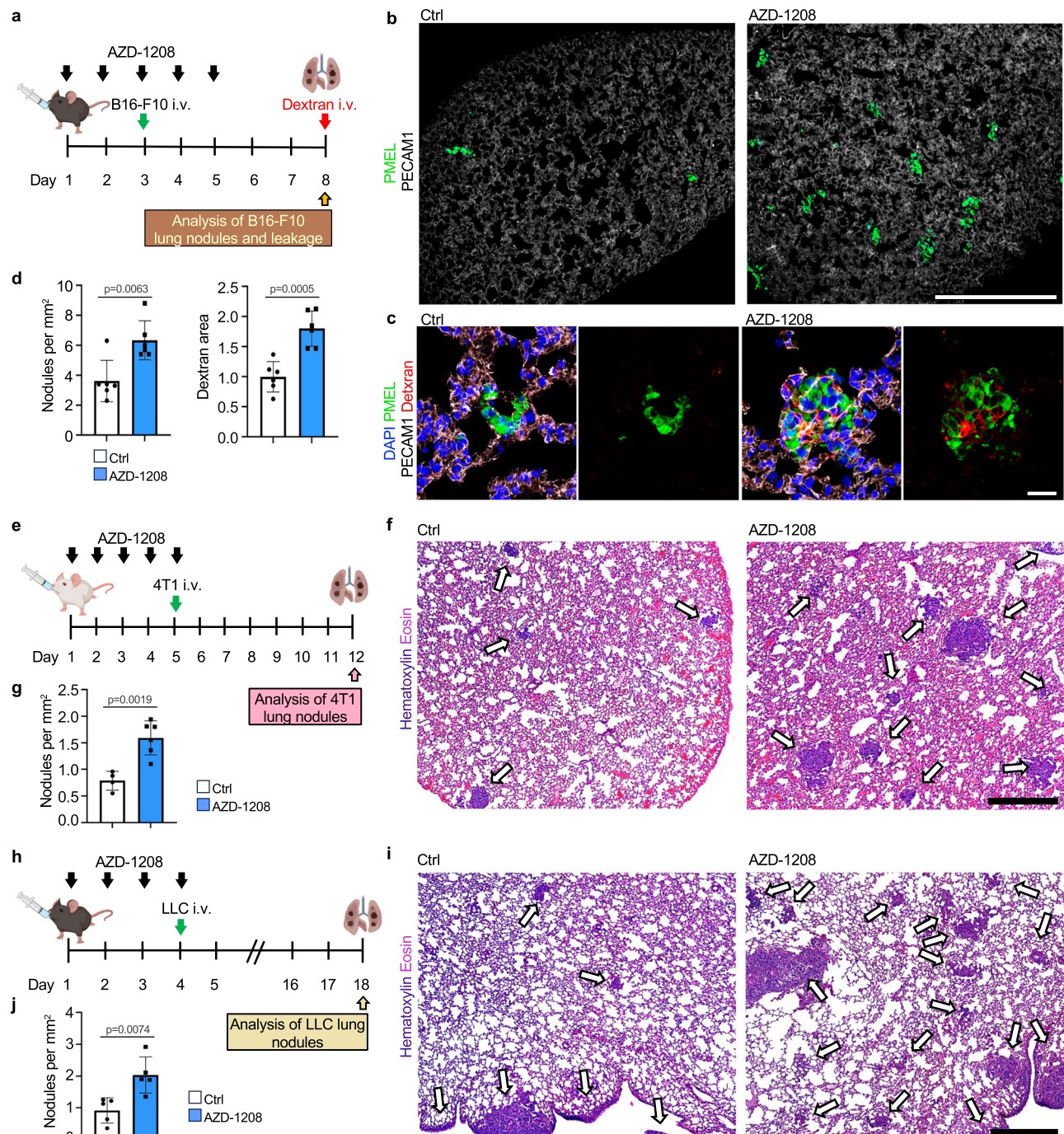

**Fig. 7 | PIM inhibitor pre-treatment increases vascular leakage and tumor cell colonization. a**–**d** PIM inhibitor AZD-1208 (30 mg kg$^{-1}$) or vehicle (Ctrl) was orally administered daily for 5 days. On day 3, B16-F10 melanoma cells were injected intravenously (i.v.) and 70 kDa fluorescent dextran was injected for the last 10 min on day 8 (**a**). Shown are representative confocal images (**b**, **c**) and number of metastatic nodules and dextran area quantified from lung sections (**d**). $n = 6$ mice per group. **e**–**g** AZD-1208 or vehicle (Ctrl) were administered orally daily for 5 days. On day 5, 4T1 cells were i.v. injected into BALB/c female mice (**e**). Representative images (**f**) and quantification of metastases from hematoxylin & eosin (H&E) stained lung sections from AZD-1208 and vehicle treated mouse lungs on d 12 (**g**). $n = 4$ Ctrl mice; $n = 6$ mice in AZD-1208 group (**f**, **g**) **h**–**j** AZD-1208 was administered daily for 4 days prior to LLC i.v. injection into C57BL/6 male mice (**h**). Shown are representative images of HE stained lung sections (**i**) and quantification of metastases from AZD-1208 and vehicle (Ctrl) treated mice on d 18. $n = 5$ mice per group. (**j**) In each experiment metastases from at least four lung lobes per mouse were analyzed. Scale bar 500 μm (**b**, **f**, **i**), 25 μm (**c**). Two-tailed unpaired $t$-test (**d**, **g**, **j**). Data are presented as mean values $^{+}/-$ SD. Illustration created in Biorender.com (**a**, **e**, **h**). Source data are provided as a Source Data file.

Since the lung vascular barrier is known to form an obstacle for metastatic tumor cells[42], we investigated the effect of AZD-1208 on metastatic melanoma colonization of the lungs. We first confirmed that AZD-1208 did not affect the growth rate or apoptosis of B16-F10 in vitro using the IncuCyte® (Supplementary Fig. 11c). Also, when B16-

F10 melanoma cells were i.v. injected and let to colonize the lungs for 3 days, followed by AZD-1208 treatment daily for 5 days, no change in melanoma growth was detected between AZD-1208 and vehicle treatments (Supplementary Fig. 11d–f). However, when AZD-1208 was administered for 3 days prior and 2 days after melanoma cell injection,

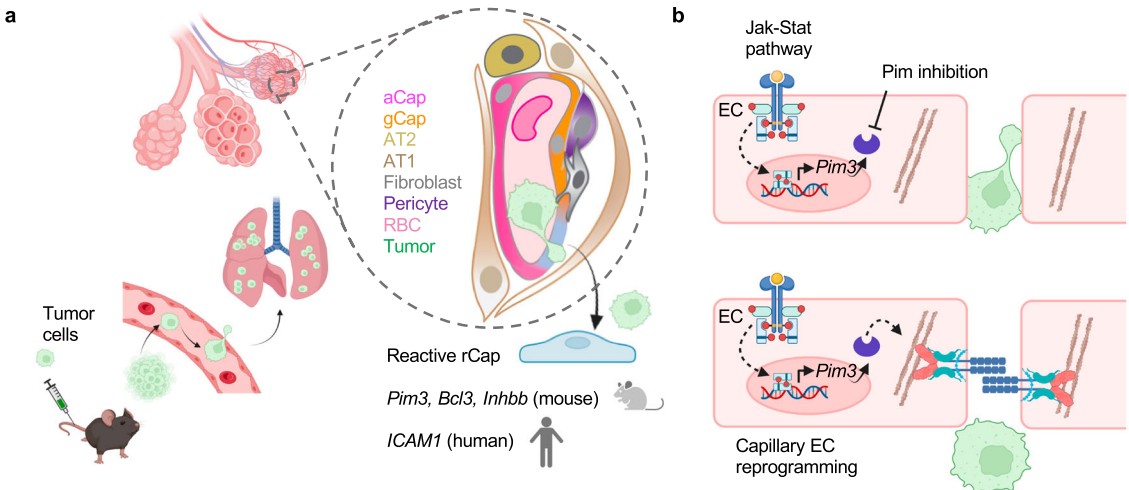

**Fig. 8 | Reactive capillary ECs in the lungs. a** Reactive capillary ECs (rCap) present in the lungs are increased in response to infiltrating tumor cells. rCap markers such as *Pim3*, *Bcl3* and *Inhbb* mark the cancer reactive niche ECs in mouse, while rCaps in the human lungs express *ICAM1*. **b** JAK-STAT pathway target *Pim3* enforces EC-EC junctions, while PIM kinase inhibition increases vascular leakage and metastatic colonization of the lungs. Illustration created in Biorender.com (**a**, **b**).

the time of metastatic colonization, AZD-1208 increased the number of metastatic nodules and leakage in the lungs (Fig. 7a–d). These results suggest that PIM inhibition increases vascular leakage and metastatic colonization, without affecting melanoma growth after formation of metastatic nodules.

To investigate the broader impact of PIM inhibition on tumor metastasis and leakage, we used the 4T1 and LLC tumor cell lines, which expressed *Pim3* and 4T1 additionally high levels of *Pim1* (Supplementary Fig. 12a). 4T1 was also highly sensitive to AZD-1208 induced apoptosis (Supplementary Fig. 11c, Supplementary Fig. 12b). Therefore, to circumvent the growth inhibition, we administered AZD-1208 in mice for 4–5 days prior to 4T1 and LLC injection i.v. and monitored metastases after 7 and 14 days post tumor cell injection, respectively. AZD-1208 increased the number of metastatic nodules in both models (Fig. 7e–j). Moreover, administration of AZD-1208 in mice with orthotopic 4T1 tumors for 7 days increased lung leakage by day 20, before visible lung metastases (Supplementary Fig. 12c–f), while liver metastasis were decreased, likely due to the AZD-1208 induced 4T1 cell death (Supplementary Fig. 12g, h). Thus, our results suggest that cancer reactive EC marker, PIM3, protects the EC barrier during cancer cell colonization, while PIM inhibition increases vascular leakage and lung colonization by different cancer types (Fig. 8a, b).

## Discussion

Blood circulation serves as a gateway for CTCs, which, upon entering the lungs, become entrapped in alveolar capillaries leading to lung metastases through interactions with ECs. Here, we defined the immediate transcriptional reprogramming of the lung vasculature in response to the entrapped CTCs using scRNA-Seq. We discovered a reactive capillary EC identity in mouse lungs (rCap), which was expanded after six hours of melanoma cell arrival through reprogramming from general capillary ECs (gCap), with stem/progenitor cell function in the alveolar-capillary bed[9]. The upregulation of rCap marker genes was temporally regulated in a tumor-dependent manner and enriched near infiltrated cancer cells in the lungs, supporting a local rather than systemic response of the rCaps. Thus, our model captured the potential founder ECs of the vascular metastatic niche of the lung and their initial transcriptional changes after tumor cell arrival.

Cross-species comparison of our mouse lung EC dataset with ECs from the Human Lung Cell Atlas revealed the rCap cluster also in the human lungs, with a high proportion of shared markers between mouse and human rCap. *ICAM1* was identified as a marker of human rCap and it, along with other rCap markers, was also expressed by

intermediate aCap (Int-aCap), which are reported to be enriched in human adenocarcinoma compared to healthy lungs[9]. Whereas post-capillary venules typically upregulate ICAM1 expression to mediate leukocyte transendothelial migration (TEM), leukocyte emigration occurs in lung capillaries, which have been reported to constitutively express ICAM1[43]. Our results therefore suggest that the *ICAM1*⁺ rCaps represent potential sites of leukocyte TEM in human lungs. Moreover, homotypic interactions between ICAM1 on endothelial and on breast cancer cells were found to facilitate breast cancer metastasis in the lungs[44]. Therefore, it is interesting to speculate that the *ICAM1*⁺ rCaps represent favorable sites of tumor cell extravasation and metastasis in the lungs.

Comparing our mouse lung EC data with ECs from mouse lung tumors revealed enrichment of the rCap signature in the tumor post-capillary venules, further supporting the rCaps' role in cell trafficking. However, rCap markers were also expressed by tumor capillary ECs (tCap) and breach ECs, which are suggested to breach the basal lamina allowing tip ECs to lead vessel growth[23]. Interestingly, rCap expressed the tip and breach EC marker *Col4a2*[23] as well as tip cell markers *Dll4* and *Pfkfb3*[26,34]. The enrichment of rCap markers in the mouse tumor vasculature suggests overlapping endothelial gene signatures during both metastasis and primary tumor development and potential initial reprogramming of rCaps toward an angiogenic tumor vasculature by infiltrating tumor cells.

Although our initial analysis was based on i.v. injected B16-F10 melanoma cells in mice, which allowed investigation of EC repro-gramming independently of a primary tumor, and in the absence of increased immune cell trafficking into the lungs, we confirmed the rCap marker upregulation using the spontaneous subcutaneous B16-F10 and orthotopic mammary carcinoma 4T1 metastasis models. Upregulation of rCap markers in the lung ECs showed a temporal dependency on increased CTCs in these models. In line with melanoma cells infiltrating the lungs but not the liver after tail vein injection, rCap markers were upregulated in the lungs but not the liver; however, they were upregulated in 4T1 liver metastases, suggesting that tumor cells reprogram the metastatic niche in these tissues. The upregulation of rCap markers in two different organs and by three tumor cell lines in vitro or in vivo suggest common or partially overlapping tran-scriptomic signatures of the organ-specific metastatic vascular niches. This conclusion is consistent with previous reports where a similar organ-independent EC response to metastasis was detected[37]. How-ever, also organ-specific EC factors regulating metastasis have been reported[5,45].

Gene set enrichment analysis revealed the rapid transcriptional activation of the JAK-STAT pathway in mouse lung ECs after melanoma cell arrival and in HUVECs in response to tumor cell-secreted factors, aligning with its established role in metastasis[35–37]. Although the upstream signals remain to be identified, the downstream targets of the JAK-STAT pathway including the serine/threonine kinase *Pim3* and *Bcl3* were identified as rCap markers. We focused on PIM3, since limited data is available on the role of PIM3 in ECs[46], whereas cancer cell expressed PIM kinases are well-known oncogenes, which promote cancer cell motility, metabolism, proliferation as well as metastasis and tumor angiogenesis[11,47]. Moreover, the unique kinase domain structure has made PIM kinases highly attractive targets for anti-cancer drug development[11,47].

Here, we utilized the selective ATP-competitive pan-PIM kinase inhibitor AZD-1208, which inhibits all PIM kinases, including PIM1 and PIM3 expressed in lung ECs[15]. Unexpectedly, AZD-1208 treatment before melanoma cell injection in mice caused lung vascular leakage, gaps in alveolar EC junctions, and increased lung metastatic nodules, but had little effect on the growth of existing melanoma metastases during the observed time window. AZD-1208 increased metastatic colonization of also AZD-1208 sensitive 4T1 cancer cells, indicating that this effect was independent of the cells' sensitivity to AZD-1208-induced cell death. These results support the intriguing hypothesis that the effect of PIM inhibition depends on the addiction of tumor cells to PIM-mediated cell survival, whereas PIM inhibition independently of a tumor type, increases lung EC permeability, facilitating metastasis. However, the potential consequences of PIM inhibition on other vascular beds remain to be investigated.

In cultured ECs, PIM inhibition and PIM3 gene silencing impaired barrier function and reduced surface CDH5 at EC-EC junctions, along with α-, β-, and δ-catenins that link CDH5 to the actin cytoskeleton. Since PIM1 was expressed at low level in cultured ECs, these results further support the role for PIM3 in stabilizing EC junction integrity. However, the upstream signals inducing PIM3 transcription and the downstream signals involved in EC barrier stabilization, including the potential role of PIM3-mediated regulation of NOTCH[14,26,48,49], remain subjects for further study.

In conclusion, our results reveal a reactive capillary EC identity in mouse and human lungs, demonstrating the plasticity of the pulmonary endothelium in response to tumor cells. Whereas tumor-induced changes in EC permeability are known to promote metastasis[50] our results suggest an anti-metastatic role for PIM3 in maintaining adherens junction integrity. The barrier-protective role of PIM3 and the effect of PIM inhibitor on vascular permeability may partly explain the limited efficacy of PIM inhibitors in cancer trials[17,18].

## Methods

### Mouse experiments

All experimental procedures involving mice were approved by the Project Authorization Board, Regional State Administrative Agency for Southern Finland and performed under the license ESAVI/15852/2022. Male C57BL/6JRj mice were used. rCap marker expression and the effect of AZD-1208 on vascular leakage and metastasis were confirmed in female BALB/cJRj mice. Mice were used between 8 and 12 weeks of age, both strains from Janvier labs (Le Genest-Saint-Isle, France). Mice were housed in individually ventilated cages with enrichment materials in a specific pathogen-free facility following the guidelines by the Federation of European Laboratory Animal Science Associations at $21 \pm 1\,°C$ and relative humidity between $55 \pm 10\%$ under 12 h dark/light cycle.

### Cell culture and reagents

The mouse skin melanoma B16-F10, mammary gland carcinoma cell line 4T1 and Lewis Lung Carcinoma cell line LL/2 (LLC1) (American Type Culture Collection, ATCC®CRL-6475™, CRL-2539™ and CRL-1642™ Manassas, VA, USA) and B16-F10-eGFP-Puro (CL053, Imanis Life

Sciences, Rochester, MN, USA) were cultured in Dulbecco's modified Eagle medium (Lonza, Basel, Switzerland) supplemented with 10% fetal bovine serum, $50\,\mu$ ml$^{-1}$ of penicillin, $50\,\mu g$ ml$^{-1}$ of streptomycin and 2 mM L-glutamine. CellTracker™ Green CMFDA Dye (C2925, Thermo-Fisher Scientific, Waltham, MA, USA) was used to label B16-F10. Human Umbilical Vein ECs (HUVEC) (Cell Applications, Inc., San Diego, CA, USA) and Human Dermal Blood Microvascular Endothelial Cells (a.k.a. BEC, HMVEC-dBl-Neo, cc-2813, Lonza) were maintained in EBM™−2 EC Growth Basal Medium-2 supplemented with EGM™-2 MV Micro-vascular Endothelial SingleQuots™ Kit (Lonza). Pan-PIM inhibitor AZD-1208 (Sigma-Aldrich, St.Louis, MO, USA) was diluted in DMSO (0.1% final concentration). Primers and antibodies are listed in Supplementary Tables 2–4.

### Mouse tumor and metastasis models

PBS (100 μl), B16-F10 (200,000 or 400,000 cells in 100 μl) or LLC cell suspension (200,000 cells in 100 μl) in PBS were injected intravenously (i.v.) via the tail vein or subcutaneously (s.c.) in 8–10-week-old C57BL/6JRj male mice (Janvier labs), and lungs, blood samples or subcutaneous tumors, were collected at indicated time points for analysis[51]. For orthotopic inoculations of mammary carcinoma, 4T1 cells (200,000 in 100 μl PBS) were injected to the fourth mammary fat pad of BALB/cJRj female mice (Janvier labs). Tumor width and length was determined by caliper, after which tumor volume was estimated by the formula: Volume = (width)$^2$ × length/2. The maximal tumor size diameter of 15 mm was followed, and experiments were terminated accordingly. AZD-1208 (30 mg kg$^{-1}$ in 100 μl of 10% DMSO, 40% PEG300, 5% Tween-80 in PBS (vehicle)) or the vehicle alone was administered through oral gavage once-daily for 3-7 consecutive days[52]. For analysis of leakage, 1 mg of Dextran (Texas Red™, 70,000 MW, Lysine Fixable, Thermo Fischer Scientific) in 100 μl of PBS per mouse was i.v. injected and allowed to circulate for 10 min followed by sequential perfusion with PBS and 1% paraformahdehyde (PFA) in PBS via the left ventricle[53]. Organs and tumors were photographed upon dissection.

### shRNA silencing

Lentiviral vectors were produced by transfecting HEK293FT cells with shPIM3 clones TRCN0000037414: CGCCTGTCAGAAGATGAACAT and TRCN0000037416: CGTGCTTCTCTACGATATGGT (Sigma-Aldrich) from the TRC1 library together with packaging plasmids pCMVg and pCMVdelta8.9. MISSION® pLKO.1-puro Empty Vector Control Plasmid DNA (SHC001) was used as a control (shScr). ECs were transduced with shRNA lentiviral vectors in the presence of 0.1% Polybrene (Sigma-Aldrich) for 5 h, continued o/n after adding 2× culture medium and replaced with fresh complete medium and analyzed after 48 h.

### Tumor-EC coculture

200,000 HUVEC cells were plated on 12 well plates and let to form a confluent monolayer. The next day, 50,000–100,000 B16-F10-eGFP-Puro cells were added for 16 h. The cultures were imaged using EVOS FL (Therfmo Fisher Scientific). ECs and tumor cells were sorted using BD Influx™ (BD Biosciences, Franklin Lakes, NJ, USA) or Sony SH800Z (Sony Biotechnology, San Jose, CA, USA) for RNA isolation. For culture in cancer cell conditioned medium, 50,000 HUVECs were plated on 24 well plates for 24 h, and subsequently incubated in conditioned medium supernatant derived from HUVEC, B16-F10, 4T1 or LLC1 cultures or empty plates as a control, all maintained in EBM-2 medium with supplements (Lonza) for 48 h.

### Cell confluency and apoptosis assay

Cell confluency and apoptosis were analyzed using Incucyte S3 (Essen Bioscience, Ann Arbor, MI, USA) with IncuCyte® Caspase-3/7 Green Apoptosis Assay Reagent 1:1000 (Essen Biosciences). Apoptosis counts were normalized to confluence.

## Electrical cell impedance assay (ECIS)

26,000 BECs or 20,000 HUVECs were plated on E-plates (Applied Biophysics. Inc., Troy, NY, USA) and grown with continuous impedance measurement using ECIS Z Theta and analyzed using ECIS software (v.1.2.252 O PC, 10 February 2018, Applied Biophysics. Inc., Troy, NY, USA). For cell stimulations, conditioned media was diluted 1:1 using basal medium. After reaching the plateau, the treatments were applied in 10 µl total volume.

## Immunofluorescence staining of cell coverslips

Cells were plated on cover glasses, grown until confluent for 24–48 h, starved in conditioned medium diluted with basal medium 1:1 for 2 h, treated in the same medium, fixed for 10–15 min in 4% PFA-PBS at room temperature (RT), washed 3 × 5 min with PBS and permeabilized for 5 min with 0.1% Triton X-100. Cells on coverslips were blocked for 10 min in 1% BSA-PBS, incubated for 30 min with primary antibodies in 1% BSA-PBS, washed with PBS, blocked, incubated for 30 min with secondary antibodies, washed with PBS and mounted with DAPI-mounting medium (ab104139, ThermoFisher Scientific)[53]. For cell surface staining, ECs were blocked on ice in 1% BSA-EBM2 for 10 min, stained using primary antibodies in 1% BSA-EBM for 30 min, washed using BSA-EBM2 and subsequently using DPBS (Gibco/Thermo Fisher Scientific), and fixed using 4% PFA for 20 min, washed and mounted.

## RNA isolation

Blood was collected through the hearth using MiniCollect® TUBEs (1 ml K3E K3EDTA, Greiner Bio-One, Kremsmünster, Austria), and RNA was isolated using NucleoSpin® RNA Blood (Macherey-Nagel, Düren, Germany). RNA from cultured cells, or cells collected from mouse tissues, was isolated using NucleoSpin RNA kit (Macherey-Nagel).

## Isolation of ECs from mouse lungs and liver

For isolation of cells from mouse lungs and liver, single cell suspensions were prepared by mincing tissues on ice and digesting for 1 h at 37 °C using 1 mg ml$^{-1}$ collagenase Type I (Gibco/Thermo Fisher Scientific), 0.1% BSA and 7.5 µl ml$^{-1}$ of DNAseI (Roche, Basel, Switzerland) in DPBS. Thereafter, samples were vortexed and passed through a 40 or 70 µm Nylon cell strainer (Corning Life Sciences or Thermo Fisher Scientific), the cells were centrifuged and resuspended into red blood cell (RBC) lysis buffer (155 mM NH$_4$Cl, 12 mM NaHCO$_3$, 0.1 mM EDTA) for 2 min, centrifuged and washed using FACS buffer (0.1% BSA, 2 mm EDTA in PBS). For liver samples, the RBC lysis was performed twice. ECs were stained using anti-CD31-PE-conjugated (a.k.a. PECAM1) and anti-CD45-APC-conjugated (a.k.a. PTPRC) antibody (Supplementary Table 2) in FACS buffer for 30 min, washed twice with FACS buffer and sorted as described for preparation of scRNA-Seq samples.

## RT-qPCR

RNA reverse-transcription into cDNA was performed using SensiFAST cDNA Synthesis Kit (Bioline, London, UK). For real-time PCR the DyNAmo HS SYBR Green master mix (Thermo Scientific) and the BIO-RAD C1000 Thermal cycler (Bio-Rad Laboratories, Hercules, CA, USA) were used. Samples were normalized to mouse *Hprt* or human *HPRT* and the fold changes were calculated using the comparative CT (threshold cycle) method. The primers (Sigma-Aldrich) are listed in Supplementary Table 4.

## Western blotting

Total lysates of mouse lung tissue were prepared in RIPA buffer (50 mM Tris-HCl pH 7.4, 1% NP-40, 0.25% Na-deoxycholate, 150 mM NaCl, 1 mM EDTA) containing protease and phosphatase inhibitors (Protease inhibitor cocktail tablets (Roche#04693159001), 1 mM Na$_3$VO$_4$ and 1 mM NaF) using PowerLyzer 24 Homogenizer (110/220 V) (Qiagen, Hilden, Germany). ECs were captured from collagenase-dissociated cell suspensions from the lungs using PECAM1 antibody

and Dynabeads Protein G beads (Thermo Fisher Scientific). 10 µg of total protein, measured using Pierce™ BCA Protein Assay (Thermo Scientific) and prepared in reducing Laemmli sample buffer, was loaded per well, separated on Novex Wedge Well 8% or 4–20% TRIS-glycine polyacrylamide gels (Invitrogen) and transferred to a methanol-activated PVDF membrane (Merck Millipore). The membrane was blocked for 30–60 min in 0.05% Tween-20−TBS containing 5% BSA and probed with primary antibodies (Supplementary Table 2) at +4 °C overnight. Secondary antibody (Supplementary Table 3) incubation was performed for 30 min at RT[53]. Western blot signal was imaged using Odyssey CLx near-infrared fluorescence imaging system (LI-COR Biosciences, Lincoln, NE, USA) or Azure 500 imaging system (Azure Biosystems, Dublin, CA, 94568 USA), and quantified using Fiji (1.48 s, Fiji, Wayne Rashband, National Institutes of Health, Bethesda, MD, USA).

## Immunohistochemistry of frozen tissue sections

Tissues were fixed in 4% PFA o/n at +4 °C, washed in PBS (5 × 15 min) and treated with 30% sucrose o/n at +4 °C. Samples were casted into Tissue-Tek Cryomolds (Sakura Finetek, Torrance, CA, USA). Prior to staining, 7–14 µm sections were thawn, rehydrated in 0.3% Triton-PBS (5 min) and blocked in 5% BSA, 5% Donkey serum in 0.1% Triton in PBS (1 h at RT). Incubations using primary (Supplementary Table 2) antibodies were performed in the blocking buffer o/n at +4 °C, while secondary antibodies (Supplementary Table 3) were incubated at RT for 1 h, followed by washing 4 × 10 min with 0.3% Triton-PBS. Sections were fixed with 4% PFA for 10 min, washed 2 × 5 min in PBS and mounted with DAPI-mounting medium (Thermo Fisher Scientific).

## Staining of thick lung sections

Mouse lungs were infused with 2% low melting point agarose (Sigma-Aldrich) through the trachea. Solidified tissues were rinsed with PBS and fixed with 4% PFA for 15 min at RT and for 2 h at +4 °C and washed with PBS. Samples were cast in 2% agarose and let solidify for 1 h on ice. 150 µm sections were permeabilized and blocked for 1 h at RT in Dulbecco´s donkey Immunomix in 5% donkey serum, 0.2% BSA, 0.3% Triton-X, 0.5% Sodium Azide in DPBS. Samples were stained using primary antibodies (Supplementary Table 2) for 48 h at +4 °C, washed 8 × 15 min in 0.3 % Triton-X-100 in PBS, stained using secondary antibodies (Supplementary Table 3) o/n at RT, washed as above, fixed in 4% PFA for 10 min at RT, washed with PBS 2 × 10 min, and mounted with DAPI mounting medium (Abcam, Cambridge, UK).

## HE staining of paraffin embedded tissue sections

For H&E staining, lungs and liver were fixed in 4% PFA o/n. Tissues were then washed in PBS and embedded in paraffin. 4.5 µm sections were cut from paraffin embedded blocks, deparaffinized in xylene, rehydrated through graded ethanol series (100%, 95%, 70% and 50%), washed with H$_2$O, stained with Hematoxylin (Mayer's hemalum solution) and Eosin (Eosin Y solution with 70% ethanol and 5 mL glacial acetic acid) (H&E), dehydrated through 50%, 96% and 100% ethanol followed by xylene treatment[54].

## RNA in situ hybridization

Tissues were fixed in formalin for 48 h, dehydrated, embedded in paraffin and sectioned at 4.5 µm. RNAScope Multiplex Fluorescent Reagent Kit (Advanced Cell Diagnostics; Nevark, CA, USA) was used for ISH. Specifically, the sections were incubated for 1 h at 60 °C, deparaffinized 2 × 5 min with xylene and 2 × 1 min with 100% EtOH, air dried, treated with hydrogen peroxide for 10 min at RT, washed with H$_2$O, incubated in target retrieval buffer for 15 min at 98 °C, washed with H$_2$O and 100% EtOH, air dried, treated using Protease Plus for 15 min at 40 °C and washed twice with H$_2$O. RNAScope® probes were hybridized for 2 h at 40 °C, after which HRP signal was developed for each C1, C2 or C3 probe. The following RNAscope® probes and dyes (1:750) were

used: Mm-*Pim3* (875481) with opal 620/Texas Red dye (FP1495001KT), Mm-*Inhbb*-C2 (475271-C2) or Mm-*Bcl3*-C2 (528431-C2) with Opal 690/Cy5.5 dye (FP1497001KT) and Mm-*Pecam1*-C3/Cy3 (316721-C3) with opal 570 dye (FP1488001KT) (Akoy Biosciences, Marlborough, MA, USA). After RNAscope Multiplex Fluorescent assays, the sections were blocked with 2.5% normal horse serum (S2000, Vector laboratories, Burlingame, CA, USA) diluted 1:20 in Dako antibody diluent (S080983, Agilent, Santa Clara, CA, USA) for 30 min at RT and counterstained with PMEL antibody for o/n at +4 °C and Alexa Fluor 488-conjugated anti-rabbit secondary antibody for 1 h at RT (Supplementary Tables 2, 3). Antibodies were diluted in Dako antibody diluent. Prior to mounting, DAPI counterstain was added for 30 s, after which samples were mounted with ProLong Gold Antifade Mountant (Invitrogen, Waltham, MA, USA). For IHC, serial paraffin sections were first deparaffinized as above, followed by antigen retrieval using low pH Na-citrate buffer (pH 6.0, 100 mM Sodium Citrate, 0.5% Tween20) for 5 min in 750 W and 10 min in 450 W microwave. The slides were then blocked in blocking buffer (5% BSA, 5% Donkey serum in 0.1% Triton in PBS) for 1 h at RT, incubated with Podocalyxin and αSMA primary antibodies (Supplementary Table 2) diluted in the blocking buffer o/n at +4 °C, washed and incubated with secondary antibodies (Supplementary Table 3) in the blocking buffer for 1 h at RT, followed by washing 4 × 10 min with 0.3% Triton-PBS. Sections were then fixed with 4% PFA for 10 min, washed 2 × 5 min in PBS and mounted with DAPI-mounting medium (Thermo Fisher Scientific).

## Microscopy and image analysis

Tissues sections were imaged using Axio Imager.Z2 upright epifluorescence wide-field microscope with Zen 3.1 (blue edition) software (Zeiss, Oberkochen, Germany), Zeiss LSM780 inverted confocal microscope, high-speed spinning-disk confocal Andor Dragonfly 505 (Oxford Instruments, Abingdon, UK) and Pannoramic 250 FLASH II Digital Slide Scanner (3D Histech, Budapest, Hungary). Images were either directly opened in Fiji for analysis or first exported by Pannoramic Viewer/ Case/ Slide Viewer (3D Histech) or Imaris 10.0.0 Cell Imaging software (Oxford instruments). Quantification of B16-F10 cells, dextran leakage, macrophages and neutrophils in the mouse lungs was performed by analyzing the signal area after thresholding of CellTracker green or PMEL, dextran, CD68 or S100A8, respectively, in the whole lung section, and normalized to total area. For analysis of RNA ISH samples, 7 representative ~0.01 mm$^2$ areas were analyzed for positive signal area per nuclear area from control and metastatic lung. For analysis of signal in the metastatic lungs, *Pim3*, *Bcl3* and *Inhbb* signals were similarly quantified from $7 + 7$ representative areas per lobe proximal to (<50 μm distance from cancer cells) and distant to cancer cells (>100 μm distance to cancer cells). Colocalization of RNA ISH was analyzed from three representative areas per sample (~0.015 μm$^2$) utilizing ImageJ/Fiji Cell counter to manually mark the cells based on positive signal on top of a detectable nucleus. In vitro CDH5 signal area was calculated from five maximum intensity projection images per sample after thresholding and results were normalized to the number of nuclei. For analysis of junctional catenin signal, nuclear signal was manually ruled out. In vivo, CDH5 signal and area analysis were normalized to collagen IV area. In total, Z-stacks of confocal microscopic images from three representative areas per mouse lung were imaged from 150 μm thick sections and analyses were performed on stack (81 layers) montages. CDH5 gaps were calculated manually from nine representative equal-sized areas per mouse lung lobe. Size of the analyzed field was determined in the 3D stack based on area x stack depth (144,000 μm$^3$) and results expressed per 1 mm$^3$. Imaris (version 9.8) surface masking tool was utilized for visualization of CDH5. All intensity analyses were performed using original images using similar brightness and contrast adjustments for all compared images, captured using the same exposure and intensity settings.

## Preparation of single cell suspension for scRNASeq

200,000 (experiments 1 and 2) or 400,000 (experiment 3) CellTracker™ Green labeled B16-F10 cells in PBS (100 μl) or vehicle only, were i.v injected to two (experiment 1) or three (experiment 2 and 3) mice each for 6 or 30 h. Single-cell suspension was prepared as described above for isolation of ECs from mouse lungs. The cells in suspension were stained using CD31-PE (a.k.a. PECAM1) and CD45-APC (a.k.a. PTPRC) antibodies. PECAM1$^+$ cells (experiment 1) and PECAM1$^+$ PTPRC$^-$ cells (experiments 2 and 3) and CellTracker™ Green labeled B16-F10 cells (experiments 2 and 3) were collected using BD Influx™ (build 1.2.0.108) with BD FACS™ Software version 1.2.0.142 (BD Biosciences, Franklin Lakes, NJ, USA) during ~10 min per sample (each sample consisting of cells pooled from two (experiment 1) or three (experiment 2 and 3) similarly treated mice as mentioned above). After sorting, viable cell density varied from $3.06 \times 10^5$ to $4.07 \times 10^5$ cells ml$^{-1}$, while viability varied from 87.1% to 95%.

## Single cell RNA sequencing

As detailed in Fig. 1a, three independent experiments, consisting of control and metastatic samples (each sample consisting of cells pooled from two to three mice, as explained above) were performed. Following cell sorting, 0.04% BSA was added to the sorted single-cell suspensions, after which cell viability and singlet vs duplet percentages in the samples were measured (LUNA-FL™, Dual Fluorescence Cell Counter, Logos Biosystems, Annandale, VA, USA). Thereafter, scRNAseq libraries were prepared using the Chromium Single Cell 3′ Reagent Kits v2 (10× Genomics; Pleasantom; CA, USA). The cell recovery aim for each library was 8000–10,000 cells. Libraries were sequenced in NovaSeq 6000 system (Illumina, San Diego, CA, USA) using S4 flow cell with read length of 28 + 8 + 89 followed by multiplexing and mapping to the mouse genome (build mm10) using CellRanger (10×Genomics, version 2.1.1.). The CellRanger software (10×Genomics) was used for generation of expression matrices. Data were aggregated using Cell Ranger software, while processing of the raw data was continued in R versions ranging from 3.6.2. to 4.4.0 (www.r-project.org) by utilizing Rstudio 2021.09.2 + 382 (rstudio.com) and the package Seurat v5 according to instructions by the Satija laboratory[55], as detailed below and in Supplementary Note 1.

**Filtering, normalization, and doublet identification.** Quality control of all scRNASeq datasets was performed by selecting genes expressed in more than 3 cells (min.cells = 3) and cells expressing 300 genes (min.features = 300) in which less than 20% of unique molecular identifiers (UMIs) were derived from the mitochondrial genome. Each sample was individually clustered for analysis of doublets. For data processing the following functions were utilized: SCTransform (default settings), RunPCA, RunUMAP, FindNeighbors and FindClusters (resolution 0.5). Multiplet Rate (%) estimation guidelines from 10×Genomics were used for adjusting the expected doublet percentage (https://kb.10xgenomics.com/hc/en-us/articles/360001378811-What-is-the-maximum-number-of-cells-that-can-be-profiled). DoubletFinder was used to identify the doublets (pN = 0.25). The pK value was adjusted per sample.

**Integration and cluster identification.** After identification of doublets, the samples were merged. For data integration cells expressing 1000–5000 genes were selected. SCTransform and RPCA integration method were used in Seurat v5 workflow. Clusters were identified at resolution 0.8 and 30 PCs. Cluster-specific marker lists and previously reported markers[21–23] were used for identification of cell type identity. Cluster markers were identified by PrepSCTFindMarkers and FindAllMarkers functions. Additional confirmation of cell type identity was performed based on cluster marker analysis by Ma´ayan Lab Enrichr tool (https://maayanlab.cloud/Enrichr/). Low-quality ECs and non-ECs were excluded from further analysis. Low quality was determined

based on low average count (~4000) and nfeature (<2000) values or based on elevated doublet percentage (>40%). Filtered EC and melanoma cell subset was selected for reclustering and final data analysis. SCT normalization and RPCA integration were used, followed by clustering at resolution 0.12 with 24 PCs. Seurat[55], DoubletFinder[56], Maftools[57] and other packages are listed in Supplementary Data 6. Three ambiguous cells with non-specific markers but clustering as melanoma cells were manually removed from the healthy control sample.

**Data visualization.** Clusters were visualized using UMAP plot. Gene expression was visualized using dot plot, Feature plot, Custom Feature plot (scCustomize; https://github.com/samuel-marsh/scCustomize/tree/v2.1.2) and Stacked violin plot (https://github.com/ycl6/StackedVlnPlot). Fold changes in gene expression between two conditions were visualized by dot plots, created in Python. Seurat object containing the reclustered EC data from all three datasets was converted to AnnData object by SeuratDisk package for analysis by Python. Control and treatment rank_gene_groups data was obtained for each cluster subset by "sc.tl.rank_genes_groups()" function, while sc.pl.rank_genes_groups_dotplot() function was used to visualize the log2 fold changes ("values_to_plot" = 'logfoldchanges') in the combined data including all clusters. Fold change heatmaps were created in R using heatmaply utilizing ColorRampPalette.

**Compositional analysis of scRNA-Seq data.** scCODA (v0.1.9)[58], a Bayesian model based on hierarchical Dirichlet-multinomial distribution, was used to identify credible compositional differences in cell clusters between conditions. Defaults parameters were used and the reference cell cluster (Artery) was automatically selected by scCODA. Parameter inference was performed via Hamiltonian Monte Carlo sampling, the default Markov-chain Monte Carlo method in scCODA. scCODA identified a credible change (FDR < 0.05) in rCap cluster between mice treated with either PBS or B16 melanoma cells. The analyzed data contained control and metastatic lungs (6 h) from three independent experiments ($n = 3$) (experiments 1, 2, and 3).

**RNA velocity analysis.** Cell Ranger aligned bam files were used as input for velocyto CLI (v0.17) to derive the counts of spliced and unspliced reads in loom format. Next, the UMAP embedding from previously produced integrated Seurat object was converted into.h5ad file for analysis using scVelo (v0.2.5) python package[59]. Data was filtered and normalized in scVelo using default parameters. Velocity dynamics were recovered using scvelo.recover_dynamics() function. Velocities were estimated and the velocity graph constructed using scvelo.tl.velocity() function with the mode set to dynamical. Velocities were then projected on top of previously calculated UMAP embedding using scvelo.tl.velocity_embedding_stream() function.

**CellChat analysis.** EC-EC communication was compared between the control and 6 h metastatic lung samples. Basic comparison analysis of multiple datasets and analysis of datasets with different cell type compositions were utilized, due to the presence of the melanoma cluster only in the 6 h, but not control samples. For analysis of melanoma signaling pathways, the 6 h sample set from experiment 3 was visualized alone. SCT assay data with UMAP-based dimensional reduction was used as basis for CellChat cell communication analysis according to the database protocol[24]. ComplexHeatmap[60], NMF[61] and other packages are listed in Supplementary Data 6. Number and strength of interactions were visualized by netVisual_heatmap function. Significant upregulated pathways were identified by identifyOverExpressedGenes (thresh.pc 0.2, thresh.fc 0.2, thresh.p 0.001), followed by mapping onto the inferred cell-cell communications by netMappingDEG and extracting the ligand-receptor pairs for upregulated ligands in metastatic lung by subsetCommunication

(ligand.LogFC 0.1) and extractGeneSubsetFromPair. The upregulated pathways were visualized by netVisual_bubble (thresh 0.01) and using circle plots by netVisual_aggregate and netVisual_chord_gene functions.

**GSEA.** DE genes of mouse lung ECs between the control and 6 h metastatic samples were identified by FindMarkers function (FC > 0.1, min.pct > 0.1, Adjusted $p$-value < 0.05). FoldChangeFunction was utilized to create a list containing Log2 Fold Change (FC), $p$-value and adjusted $p$-value for generating FC heatmaps. DE genes with adjusted $p$ value < 0.05 were selected for GSEA, using gene set size 10–500. GSEA 4.3.3 with the MSig database were used for KEGG and Hallmark pathway analysis of the DE genes[12]. FDR 25% was kept as a limit for significantly altered pathways. Top 50 up- and downregulated DE genes are listed for all clusters in Supplementary Data 3.

**Analysis of scRNA-Seq data of mouse tumor ECs.** Mouse primary tumor EC scRNASeq data from orthotopic Lewis Lung Carinoma along with healthy lung ECs from C57BL/6 mice were obtained from Lung tumor ECTax (mouse) (https://endotheliomics.shinyapps.io/lung_ectax/)[23]. Data were filtered and normalized as described for our own data.

**Analysis of scRNA-Seq data of human lung ECs.** EC subsets from healthy human lung and from normal lung tissue adjacent to lung cancer from a carcinoma patient were obtained from the Human Lung Cell Atlas (https://hlca.ds.czbiohub.org/). EC subsets were obtained from the preprocessed data from all three Krasnow_2020 patients and all Banovich_Kropski_2020 patients, except patient VUHD68, which was excluded due to clustering separately from other samples[40,41]. Similarly to our mouse data filtering, also human data mitochondrial UMI percentage was limited to max 20%. Similar workflow was followed with the human and our mouse data. Min.cells = 3 and min.features = 300 were selected as limits prior to data integration. SCTransform was utilized to each Krasnow_2020 patient alone (donor 1: 1490; donor 2: 11686 and donor 3: 3380 cells) and to the whole Banovich_Kropski_2020 dataset (15,452 cells), after which RPCA integration was used to integrate the samples. Clusters were identified at resolution 0.45 with 25 PCs. A small subset of low-quality cells with high mt percentage and another small cluster of non-ECs were excluded, and the high-quality EC subset was reclustered with the similar workflow at resolution 0.33 with 25 PCs for final analysis. Previously identified markers along with identification of full cluster marker lists and earlier annotations[9,40,41] were utilized for cell type identification. Additionally, human data was also integrated with our mouse EC subset (control and 6 h B16-F10) to compare the rCap clustering between mouse and human. For this purpose, a down sampling of 75% was taken from the mouse data, after which nichnetr package was utilized to translate the mouse gene names to human orthologs. Again, SCTransform and RPCA integration were used. Harmony integration was also tested in Seurat5, which resulted in similar type of results, as RPCA. Final figures were created based on RPCA integration, 20 PCs and resolution 0.55.

**Gene set projections.** Different marker gene sets were projected to the scRNA-Seq datasets utilizing the UCell signature enrichment (https://carmonalab.github.io/UCell_demo/UCell_Seurat_vignette.html)[62]. AddModuleScore_UCell was utilized to calculate the enrichment scores, which were visualized by violin and feature plots as described above.

**Bulk mRNA-Seq analysis**
HUVECs were cultured for 16 h in conditioned medium originating from 48 h B16-F10 melanoma cell cultures ($n = 4$) or empty plates ($n = 4$). After RNA isolation and quality control, samples were

sequenced at Novogene (London, UK). DE genes were identified and GSEA were performed using Chipster[63] and DESeq2 (v3.18) and GSEA software (v4.3.2). Reads were aligned to GRCh38.p12 using HISAT (v2.2.1), default setting with soft clipping turned on. Mapped reads were counted using HTSeq (v2.0), reference genome GRCH38.109, default settings. Principal component analysis (PCA) was performed for sample counts. Control samples and treated samples arranged spatially in to two different groups, indicating no intra-condition variability. Differentially expressed genes were analysed using DESeq2 (v3.18), default settings with cutoff value 0.05 for Benjamini−Hochberg adjusted $p$-values. GSEA was performed using GSEA software (v4.3.2), and the Molecular Signatures Database (MSigDB) hallmark gene set collection and Kyoto Encyclopedia of genes and genomes (KEGG) pathway database. Enriched pathways with FDR < 25% are shown.

## Figure preparation
Illustrations in Figs. 1a, 2h, 4a, d, i, k, 5m, 7a, e, h, 8a, b and Supplementary Figs. 2a, 6h, 8a, 9a, f, 11a, d and 12c were created in BioRender.com (Saharinen, P. (2024) BioRender.com/p58u107). For Imaris surface masked images, Contour Find Edges Soft 50 and Directional sharpen (100%) CorelDraw tools were utilized to produce the final representative images.

## Statistical analysis
GraphPad Prism software (version 9.2.0, GraphPad Software, LLC, San Diego, CA, USA) was used for analysis of other than scRNAseq data. Two-sample comparisons were performed using a two-tailed Student's $t$-test or a Mann−Whitney U test, depending on the results of a normality test, while one-way or two-way Analysis of Variance (Anova) or Mixed-effects analysis was performed to larger datasets utilizing multiple comparison in GraphPad Prism. The number of independent experiments or mice per treatment groups are stated in the figure legend. For mouse experiments, each dot represents one mouse and for cell-based experiments, each dot represents an independent sample. In vitro experiments were performed in minimum of three times. Statistical analyses for $p$-values of scRNA-Seq data were determined in R, the default is Wilcoxon Rank-Sum Test. All R packages are listed in Supplementary Data 6.

## Reporting summary
Further information on research design is available in the Nature Portfolio Reporting Summary linked to this article.

## Data availability
The raw scRNASeq and bulk-RNA Seq data generated in this study are available in the GEO repository under accession codes GSE235394 (scRNASeq dataset 1), GSE253749 (scRNASeq dataset 2), GSE253751 (scRNASeq dataset 3) and GSE25197 (bulk mRNA-Seq). The numerical data generated in this study are provided in the Source Data file. The orthotopic Lewis Lung Carcinoma primary tumor EC scRNASeq data used in this study are available in the LungECTax database (https://endotheliomics.shinyapps.io/lung_ectax/)[23] and human EC scRNASeq data used in this study are available in the Human Lung Cell Atlas database (https://hlca.ds.czbiohub.org/)[40,41]. Biological material is available upon request. Source data are provided with this paper.

## Code availability
Applicable code generated during this study is available at GitHub: Analysis of mouse lung ECs[64] (https://github.com/nmsantio/Mouse-melanoma-lung-EC-scRNASeq); analysis of Lung EC Tax[65] (https://github.com/nmsantio/Create-Seurat-Objects-from-on-line-sources); analysis of human lung cell atlas data[66] (https://github.com/nmsantio/Integration-of-mouse-and-human-scRNASeq-data) and integration of human data with mouse ECs[67] (https://github.com/nmsantio/HLCA_scRNASeq_EC).

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

## Acknowledgements

We would like to thank K. Salo and T. Laakkonen at Wihuri Research Institute for assistance in scRNA library preparation, the following University of Helsinki units supported by HiLIFE and Biocenter Finland: FIMM Single-Cell Analytics Unit, the HiLife Flow Cytometry Unit, the Biomedicum Imaging Unit for microscopy services, the University of Helsinki Functional Genomics Unit and the Laboratory Animal Centre. A18 antibody was a kind gift from Dr. Akira Nagafuchi, Nara Medical University.

This study was supported by Wihuri Foundation (PS), European Research Council (ERC) under the EU Horizon 2020 research and innovation programme (grant agreement 773076, PS), Sigrid Jusélius Foundation (PS), Cancer Foundation Finland (PS), the Worldwide Cancer Research (18-0697, UK, PS), Academy of Finland Centre of Excellence Program (346134 PS), Helsinki Institute of Life Science (HiLIFE) Fellow Program (PS). Open access funded by Helsinki University Library.

## Author contributions

N.M.S., K.G., and P.P.K. conceived the study, designed and conducted laboratory experiments, analyzed and interpreted results. N.M.S. drafted and wrote the manuscript. K.G. and P.P.K. participated in writing the manuscript. N.M.S. analyzed and interpreted scRNASeq results. A.H., E.A., and F.S. participated in scRNASeq analysis. A.H. analyzed and interpreted bulk-RNASeq. R.K. designed experiments and participated in writing the manuscript. S.H. performed RNA ISH under the leadership of O.C. P.S. conceived and supervised the study, designed experiments, analyzed and interpreted the results, and wrote the manuscript. All authors read and approved the manuscript.

## Competing interests

The authors declare no competing interests.
