## [Peer Review File · Nature Communications]

Editorial Note: Figure 1 and 2 on page 13 and 15 of this Peer Review File have been redacted as indicated to maintain the confidentiality of unpublished data.

REVIEWER COMMENTS

Reviewer #1 (Remarks to the Author):

Summary: In this manuscript, the authors identified a novel subpopulation of endothelial cells (ECs) induced by metastatic cancer cells by performing single-cell RNA sequencing (scRNA-seq) on the syngeneic B16-F10 metastatic lung mouse model. They showed that Pim3 is a marker of the novel subpopulation (crECs) expanded in metastatic vascular niche through the JAK-STAT3-PIM3 axis. The upregulation of Pim3 in the metastatic vascular niche was also demonstrated in the spontaneous metastasis mouse models, human cancer scRNA-seq datasets, and in vitro human ECs cocultured with melanoma cells. In addition, the authors showed that PIM3 is required for the maintenance of EC tight junctions. Importantly, they demonstrated that PIM inhibition by a selective PIM inhibitor, increased vascular leakage and lung metastasis.

The present manuscript is potentially important because it provides a mechanistic insight into the clinical observations of lack of efficacy of PIM inhibitors. However, there are several major points that should be addressed to substantiate the authors' claims.

Major points:

1. Expansion of crECs in the metastatic vascular niche: Although the crECs were expanded in metastatic lung from the scRNA-seq dataset, the results are not statistically significant due to lack of replicates. The upregulation of crEC markers in lung ECs shown in Fig. 1h cannot be direct evidence of the crEC expansion by metastatic cancer cells, as these markers could be globally upregulated in all lung ECs. In Fig. 3a, the two crEC markers, *Inhbb* and *Pim3*, do not appear to be co-expressed, which argues against the validity of the proposed crEC markers and the expansion of the crECs. Furthermore, in Fig. S3c, *Pim3* was not upregulated in metastatic crECs compared to control crECs. The authors should demonstrate that the crECs are expanded in the metastatic vascular niche by flow cytometric analysis based on the crEC-specific surface markers or other methods.
2. The cellular interactions between crECs and metastatic cancer cells: The authors argue that metastatic cancer cells reprogram a specific subpopulation of ECs (crECs) with the activated JAK-STAT3-PIM3 axis. However, they did not investigate how metastatic cancer cells reprogram crECs, and this issue should be carefully addressed. For example, by performing scRNA-seq on a mixed population of ECs and metastatic cancer cells, cellular interactions between ECs and metastatic cancer cells could be inferred. In addition, bulk RNA-seq analysis on HUVEC and HUVEC+B16-F10 will be useful to identify signaling pathways that are altered by cancer cells. This combined analysis will provide a mechanistic insight into how metastatic cancer cells reprogram crECs.
3. Fig. 1b and S2a: It appears that the authors used the UMAP of Fig. S2a including non-ECs by simply plotting ECs in the same UMAP. Since the presence of non-ECs affects the global structure of the UMAP representation, the authors should replot the UMAP by excluding non-EC cells.
4. Fig. 1c: The pseudotime analysis was not properly performed to draw conclusions about the precursor-progeny relationships between EC clusters. First, the differentiation trajectories inferred by Slingshot are known to be highly dependent on clustering and dimensionality reduction algorithms. Therefore, the robustness of inferred trajectories should be validated by using other trajectory algorithms. Second, Slingshot does not predict a starting cluster, which should be provided by a user with

a biological prior knowledge. It is not clear how the authors chose a starting cluster. The directionality predicted by RNA velocity should be provided to evaluate this prediction. Finally, the main assumption of trajectory inference for scRNA-seq is that the underlying biological process is a continuous process. The EC clusters appear to be discrete clusters with no transient cells connecting these cell types. Therefore, it does not make sense to perform pseudotime analysis on all the ECs.

5. Fig. S6: It appears that Jak/Stat/Pim3 are also expressed in other cell types (e.g. epithelial cells). Instead of showing the expression of the markers for crECs, the authors should computationally map crECs onto the public human scRNA-seq datasets using their transcriptomes to show that crECs exist in the human metastatic vascular niche.

6. Fig. 7: I'm wondering if PIM inhibition increases vascular leakage and lung metastasis in the spontaneous tumor model.

Minor points:

1. Fig. 1b: All cluster numbers should be shown.
2. Fig. 2c and d: Color legends should be provided.
3. Fig. 6 b-d: The left and right bar graphs should be labeled.

Reviewer #2 (Remarks to the Author):

This is an interesting paper as it has particularly explored the early phase of metastasis and establishment of tumours in the lung, in contrast to many other papers that have looked at established metastases by single-cell sequencing. Intravenous administration of B16-F10 melanoma tumour cells was used for initial experiments and justified because of the need to look at early time points. This showed that after six hours melanoma clusters and single cells were detectable in the lungs, but this dramatically dropped after 12 hours. PECAM1-positive endothelial cells were isolated from the lungs six hours after the injection and analysed by SCS. An additional discriminator was introduced in the analysis, CDH 5 to ensure that endothelial cells were analysed.

Using the previous annotations from other groups showed the expected distribution of endothelial subtypes. However, more detailed analysis showed there were four clusters in the general capillary group.

This is a critical point and particularly figures 1B and C should be shown at higher resolution or in larger format. It is not clear how figure 1C fits into 1B. Additionally, there seems to be a small group below 4 on the diagram that is not discussed. Also, if the same more detailed analysis was done on other groups shown would more subgroups emerge?

Several marker genes for venous capillaries in this paper (Figure.1f) match marker genes of post-capillary venules found in murine LLC lung tumours in this study: Cancer Cell. 2020 Jan 13;37(1):21-36.e13. doi: 10.1016/j.ccell.2019.12.001.

Is it possible that they are the same EC but differently annotated?

On the data and figure 1, INHBB is the most significantly changed and PIM3 is amongst the least significant with a similar fold change, so the explanation for choosing PIM3 needs to be more fully explained. Does INHBB induce PIM3?

In Figure 2c missing colour scale for the green heatmaps.

Figure 2e is not particularly convincing for a difference in the control vs melanoma for VegfA interactions in crEC. Statistical comparison should be carried out to justify the comment.

Similarly, in figure 2b, the Vegf axis is lower than in the arterial capillaries with tumour and is more induced in the latter. Please discuss.

In Figure 2c Vegf pathway, it looks to me that signaling between 5 and is still stronger than between 4 [crEC] and other pathways. Also, signaling from Notch seems to be go down from 4, not up, with a general reduction. The meaning and code for the lines is not provided.

In Figure 3 the melanoma cells are shown in green and there seems to be a large distance, relatively several cell diameters, between the melanoma and INHBB and PIM3 staining. Also, interestingly the cells highlighted by the IINHBB probe seem larger than those with the PIM3 probe, although this may simply be the relative abundance of the RNA.

Several representative areas should be presented and more clarity in the explanation of the image, perhaps with arrows. A stain showing the outlines of the cells or PECAM1 protein would help understand the image far better, even if serial sections.

Overall, there is a lack of confirmation of the target genes by immunohistochemistry in the metastases.

The in vitro work with coculture of melanoma and HUVECs clearly shows the upregulation of PIM3 and INHBB after 16 hours. It would be interesting to know the time course and whether there is greater induction over 24 hours, which may mimic more the duration of the experiments in vivo?

A key question that should be answered by co-culture in separate wells is whether the initial induction of PIM3 was mediated by a soluble factor from the melanoma cells, which might be relevant for many other pathways e.g. interferon secretion. In figure S2, top 3 pathways are all interferon related, so I think the transfer of soluble factors could be a key link, needs to be tested.

I am not sure how the hypothesis is supported by the human clinical data. Although it shows these genes are upregulated, the hypothesis is that these are transient and involved reversibly in the early niche. This seems unlikely in the studies of the human metastases and should be discussed further.

The further work on PIM3 and regulation of endothelial junctions clearly shows the downregulation of CHD5 and confirms the increased vascular leakage and metastasis in the lungs.

This study is important because of its direct impact on studies and clinical research involving PIM 3 inhibitors and the risks. It is interesting though a recent phase one study in hematopoietic neoplasms showed toxicity but no evidence of activity.

Reviewer #3 (Remarks to the Author):

In this manuscript, Santio et al. characterised the diverse endothelial cell types in the lung metastatic niche after arrival of melanoma cells and revealed a novel cluster of cancer responding ECs. Their scRNA-seq results pointed to the Jak-Stat activated Pim 3 as a marker for these cells. Surprisingly, inhibition of Pim3 increased vascular leakage and metastasis by impairing the EC barrier.

The quality of the data appears very good. The new EC cluster that has been identified could have potential clinical importance. However, I have some questions and comments with regards to the analysis of the data that need to be addressed.

Major comments:

The authors used very stringent criteria for quality control, very high min of number of genes (1500) and quite low max number of genes (4500), as well as only 3% of mitochondria, was there a reason for such stringent QC metrics?

Have the authors performed statistical analysis for the differential abundance between control and metastatic lung? A method such as Milo would be helpful for this.

Fig 1c - the trajectory analysis results are a bit confusing for me, I can't really see a clear trajectory on this plot. Can the authors reproduce their results using a different method? Otherwise the claim that the novel EC cluster is derived from venous capillaries would be very speculative.

Fig 2e - is there a statistically significant difference in expression of the selected genes?

Why was a UMAP dimensionality reduction used for CellChat, instead of cluster labels or a more appropriate dimensionality reduction for clustering such as PCA?

The authors performed cell-cell communication analysis using only the sequenced EC data. It would be much more relevant to try this with cancer and other cells, using publicly available data, in order to see how the cancer cells potentially communicate with the cancer responding ECs.

Fig S3c, was there any statistically significant difference in expression of the selected genes?

From the analysis of the publicly available data, it seems that the specific markers of the cancer responding ECs are also present in endothelial cells of primary tumours, have you checked this and what is your comment on that?

Minor:

Fig 1g, 2a, it would be visually better if the annotations of clusters are used instead of numbers.

Response to Reviewer report(s)

Manuscript Santio et al. Endothelial Pim3 kinase protects the vascular barrier during lung metastasis

Reviewer comments in blue. Author response in black.

We thank the reviewers and editors for critical comments on our manuscript.

In response to the comments, we included new scRNA-Seq data from two additional independent experiments (biological replicates) performed similarly to our original experiment (n=3 in the revised manuscript). The new experiments more than triplicated the EC number in our dataset, providing a robust analysis of the novel capillary EC cluster already in healthy lung, albeit increased in metastatic lungs. Hence, we would like to rename the cluster as **reactive capillary EC (rCap)**, to better reflect its cancer-independent identity in the healthy lung. The addition of new scRNA-Seq data forced us to redo the data analysis, and therefore, some of the original figures have been replaced by new figures. Additionally, to comply with Nature Communications Figure formatting guidelines, we have relabeled some of the original figure items. The changes in figures have been outlined below, after the by point-by-point response to reviewer comments.

Reviewer #1: scRNAseq

Summary: In this manuscript, the authors identified a novel subpopulation of endothelial cells (ECs) induced by metastatic cancer cells by performing single-cell RNA sequencing (scRNA-seq) on the syngeneic B16-F10 metastatic lung mouse model. They showed that Pim3 is a marker of the novel subpopulation (crECs) expanded in metastatic vascular niche through the JAK-STAT3-PIM3 axis. The upregulation of Pim3 in the metastatic vascular niche was also demonstrated in the spontaneous metastasis mouse models, human cancer scRNA-seq datasets, and in vitro human ECs cocultured with melanoma cells. In addition, the authors showed that PIM3 is required for the maintenance of EC tight junctions. Importantly, they demonstrated that PIM inhibition by a selective PIM inhibitor, increased vascular leakage and lung metastasis.

The present manuscript is potentially important because it provides a mechanistic insight into the clinical observations of lack of efficacy of PIM inhibitors. However, there are several major points that should be addressed to substantiate the authors' claims.

Author response: We wish to thank the reviewer for constructive and positive comments, which have significantly improved our manuscript.

Major points:

1. Expansion of crECs in the metastatic vascular niche: Although the crECs were expanded in metastatic lung from the scRNA-seq dataset, the results are not statistically significant due to lack of replicates. The upregulation of crEC markers in lung ECs shown in Fig. 1h cannot be direct evidence of the crEC expansion by metastatic cancer cells, as these markers could be globally upregulated in all lung ECs. In Fig. 3a, the two crEC markers, *Inhbb* and *Pim3*, do not appear to be co-expressed, which argues against the validity of the proposed crEC markers and the expansion of the crECs. Furthermore, in Fig. S3c, *Pim3* was not upregulated in metastatic crECs compared to control crECs. The authors should demonstrate that the crECs are expanded in the metastatic vascular niche by flow cytometric analysis based on the crEC-specific surface markers or other methods.

Author response: We thank the reviewer for bringing up these crucial points. Due to the lack of antibodies against rCap (crEC) markers suitable for flow cytometry in our hands, we used other methods to provide statistical analysis of cell numbers in the rCap (crEC) cluster between control and metastatic lungs. In the revised version of the manuscript, we performed two new independent scRNA-Seq experiments (biological replicates) under identical experimental conditions as our first experiment, except for a higher number of injected cancer cells in experiment 3 (due to reasons explained below). Integration of the new scRNA-Seq data sets with our original data is shown in new Fig.1c, followed by analysis of changes in cell composition using scCODA (Büttner et al. 2021*) in new Fig.1f, which demonstrated a statistically significant increase in the rCap (crEC) cluster in metastatic vs control lung.

*Büttner, M., Ostner, J., Müller, C.L. et al. scCODA is a Bayesian model for compositional single-cell data analysis. *Nat Commun.* **12**, 6876 (2021). DOI: [10.1038/s41467-021-27150-6](https://doi.org/10.1038/s41467-021-27150-6)

The scRNA-Seq datasets are deposited in the GEO repository:

Exp1 (original dataset): GSE235394, access code: unmnswynbkbfd

Exp2 (new dataset):GSE253749, access code: sdcbkssgblpydyn

Exp3 (new dataset): GSE253751, access code: otclcwisybzuhtyf

To further demonstrate that the rCaps (crECs) are expanded in the metastatic vascular niche we performed an additional coexpression analysis of *Pim3* and *Inhbb*, along with *Pim3* and *Bcl3*, using RNAScope in situ hybridization (ISH). The new results show that *Pim3* is coexpressed with *Inhbb* in average in 16% of ECs, and with *Bcl3* in 46% of ECs, 6 h after cancer cell injection (updated Fig.3i-l). Moreover, new Supplementary Fig. 5f supports the conclusion that also *Bcl3* is increasingly expressed in the vicinity of tumor cells (within <50 mm distance from tumor cells) in comparison to ECs further away (> 100 mm distance) from tumor cells.

The ISH results are in line with results from the scRNA-Seq, where *Inhbb* is expressed in a smaller percentage of rCaps (crECs) in comparison to *Bcl3* or *Pim3* (new Fig. 1h). This differential expression of the markers may be due to the very short time (6h) of tumor-induced gene expression changes. Nevertheless, we cannot exclude the possibility that the rCap (crECs) may represent in fact more than one capillary EC identity. The subclustering is, however, beyond the scope of the current study.

We agree with the reviewer that the violin plots in the original Fig.S3c in our manuscript did not well represent the difference in *Pim3* expression between control and metastatic lungs, which was likely due to the low number of rCaps (crECs) in the control lungs (33 crECs, shown in original Fig.1e). Importantly, the new integrated scRNA-Seq data set including 3 independent experiments, allowed a more robust analysis of rCap (crEC) marker gene expression in both the control and metastatic lungs. The new dot plots in updated Fig.1h represent, in logarithmic scale, the differential gene expression of selected marker genes between control and metastatic lungs. In addition, the new violin plots in updated Fig.1i. better illustrate the enrichment of the rCap (crEC) markers, including *Pim3* between metastatic vs control lungs. Of note, the expression of certain rCap (crEC) markers, including *Pim3* is also slightly increased in venous ECs. These new results confirm that the rCap (crEC) markers are not globally upregulated in all lung EC clusters.

2. The cellular interactions between crECs and metastatic cancer cells: The authors argue that metastatic cancer cells reprogram a specific subpopulation of ECs (crECs) with the activated JAK-STAT3-PIM3 axis. However, they did not investigate how metastatic cancer cells reprogram crECs, and this issue should be carefully addressed. For example, by performing scRNA-seq on a mixed population of ECs and metastatic cancer cells, cellular interactions between ECs and metastatic cancer cells could be inferred. In addition, bulk RNA-seq analysis on HUVEC and HUVEC+B16-F10 will be useful to identify signaling pathways that are altered by cancer cells. This combined analysis will provide a mechanistic insight into how metastatic cancer cells reprogram crECs.

Author response: The reviewer raises an interesting further aspect of our study, namely the signals derived from cancer cells that regulate rCap (crEC) behavior in the metastatic niche. Although we agree that this is an important question, our primary objective was to discover the role of ECs during the initial early events in of metastatic dissemination in distant organ. Therefore, our experimental design was focused on the 6 h time point post melanoma injection via the tail vein. Since most injected melanoma cells are no longer found in the lungs at this time point (exemplified in Supplementary Fig.1b), our experimental set-up is not ideal for capturing melanoma cells for scRNA-Seq in a mixed population of ECs and metastatic cancer cells.

Nevertheless, we sorted ECs together with cancer cells from lungs 6 h after B16-F10 injection in two separate experiments (new scRNA-Seq experiments 2 and 3), and subjected both cell types to scRNA-Seq. To increase the number of cancer cells after sorting, the number of injected melanoma cells was increased to 400 000 in experiment 3 (in comparison to 200 000 cells / mouse in experiments 1 and 2). This approach enabled the capture of 91 cells in the final, filtered and integrated scRNA-Seq data set (new Supplementary Fig. 2d-e, 2g-h), and analysis of melanoma-EC communication using CellChat (new results in Supplementary Fig. 4, discussion on p. 6-7 of the revised manuscript). However, since the results are based on a relatively low number of melanoma cells, we would like to leave the functional verification of the ligand-receptor signaling pathways in regulation of the rCap (crEC) gene signature for a subsequent study.

As suggested by the reviewer, we utilized the cell culture model of HUVECs and B16-F10 to interrogate melanoma-induced pathways in ECs. In the revised version of our manuscript, we show that in addition to coculture, also the incubation of HUVECs in conditioned media from three different cancer cell cultures (B16-F10, LLC, 4T1) upregulated rCap (crEC) markers *PIM3*, *BCL3* and *INHBB*. These new results indicate that rCap (crEC) markers are regulated in HUVECs by tumor cell -secreted factors and are presented in new Fig.4k-m and Supplementary Fig. 6j.

To discover pathways induced by tumor cell -secreted factors, we cultured HUVECs for 16 h in either control or B16-F10 conditioned media and subjected HUVECs to bulk RNA-Seq. Enrichment analysis of differentially expressed (DE) genes ($p < 0.05$ and $\text{Log}_2\text{FoldChange} > 0.25$) revealed several significantly upregulated hallmarks induced by B16-F10, including metabolic, inflammatory, hypoxia and angiogenesis (new Supplementary Fig. 7). Notably, the JAK-STAT pathway was upregulated in both HUVECs and lung ECs by tumor cells. Thus, the results using *in vitro* cell culture model support results from *in vivo* implying the transcriptional activation of the JAK-STAT pathway in the metastatic lungs (as also reported by previous studies, which are referred to in our manuscript).

The new bulk RNA-Seq data are deposited in the GEO repository GSE251979, under access code utitaiggtvobtit.

3. Fig. 1b and S2a: It appears that the authors used the UMAP of Fig. S2a including non-ECs by simply plotting ECs in the same UMAP. Since the presence of non-ECs affects the global structure of the UMAP representation, the authors should replot the UMAP by excluding non-EC cells.

Author response: We thank the reviewer for this crucial aspect in our UMAP. The new UMAPs show reclustered ECs and melanoma cells after removing other cell types and are presented in updated Fig.1c and Supplementary Fig. 2h, in comparison to UMAP of all cell clusters (updated Supplementary Fig. 2g).

4. Fig. 1c: The pseudotime analysis was not properly performed to draw conclusions about the precursor-progeny relationships between EC clusters. First, the differentiation trajectories inferred by Slingshot are known to be highly dependent on clustering and dimensionality reduction algorithms. Therefore, the robustness of inferred trajectories should be validated by using other trajectory algorithms. Second, Slingshot does not predict a starting cluster, which should be provided by a user

with a biological prior knowledge. It is not clear how the authors chose a starting cluster. The directionality predicted by RNA velocity should be provided to evaluate this prediction. Finally, the main assumption of trajectory inference for scRNA-seq is that the underlying biological process is a continuous process. The EC clusters appear to be discrete clusters with no transient cells connecting these cell types. Therefore, it does not make sense to perform pseudotime analysis on all the ECs.

Author response: We agree that the Slingshot analysis is not optimal, even though we used an automatic selection of the starting cluster. Therefore, according to the reviewer's suggestion, we performed Velocity analysis, which shows the directionality of the gene expression changes based on ratios of spliced and unspliced mRNAs. The velocity analysis supports our original results that rCap (crEC) is derived from gCap and represent the future state of gCap. The new velocity analysis is presented in updated Fig. 1g and discussed on p. 6 of the revised manuscript. In the revised manuscript we have also interpreted the data more cautiously by combining the gCapA and gCapV clusters of our original manuscript into a single gCap cluster, since the validation of these subclusters, although of interest for future work, was not in the scope of this manuscript.

5. Fig. S6: It appears that Jak/Stat/Pim3 are also expressed in other cell types (e.g. epithelial cells). Instead of showing the expression of the markers for crECs, the authors should computationally map crECs onto the public human scRNA-seq datasets using their transcriptomes to show that crECs exist in the human metastatic vascular niche.

Author response: This is an interesting, yet challenging question. As described above, our analysis represents a unique data set in mice to characterize the (pre)-metastatic niche 6 hours after cancer cell arrival in the lung capillaries. Although various scRNA-Seq datasets of human cancer exist, they do not represent, to best of our knowledge, similar biological time point of the early metastatic niche.

However, to answer the reviewer's question, we used as a human reference data, the EC subset from the Human Lung Cell Atlas (HLCA, <https://hlca.ds.czbiohub.org/>). Specifically, we analyzed ECs from healthy human lung and from lung tissue adjacent to lung tumor from carcinoma patients. This was justified since rCap (crEC) were found also in the healthy mouse lungs from our new dataset with significantly increased EC numbers. Interestingly, in addition to previously identified EC identities, the rCap (crEC) cluster was identified in the human lungs, based on rCap (crEC) marker gene signature (new Supplementary Fig. 9a-e). Furthermore, we integrated the human lung EC dataset with our mouse lung EC dataset, which preserved previously annotated clusters, and confirmed the presence of the rCap (crEC) in the human lung (new Supplementary Fig. 9f-j). We also tested different integration algorithms implemented in Seurat 5, including RPCA and Harmony, and found that the clustering results remained consistent across these methods. These results therefore reveal rCap (crEC) as a novel gCap EC cluster in both mouse and human lungs.

6. Fig. 7: I'm wondering if PIM inhibition increases vascular leakage and lung metastasis in the spontaneous tumor model.

Author response: In response to the reviewer's comment, we analyzed leakage during spontaneous metastasis in Balb/c mice with orthotopic 4T1 tumors, which was increased after AZD-1208 administration, in consistent with results from C57BL/6 mice (new Supplementary Fig.12c-e).

However, 4T1 is sensitive to AZD-1208, which induced their apoptosis in vitro (new Supplementary Fig. 12b). In line with this, AZD-1208 administration during spontaneous metastasis showed a trend of decreased liver metastasis (new Supplementary Fig. 12g-h). However, 4T1 lung metastasis develop over a long time period (21-27 days), challenging metastasis experiments due to the decreased survival of 4T1 in the presence of AZD-1208. In contrast, B16-F10 melanoma is not sensitive to AZD-1208 (updated Supplementary Fig. 11c).

Therefore, to investigate whether AZD-1208 increased lung colonization of both AZD-1208 sensitive and insensitive tumor cell lines, we injected the AZD-1208 sensitive 4T1 and the Lewis lung carcinoma (LLC) i.v. in AZD-1208 pre-treated mice. The AZD-1208 pre-treatment of mice resulted in significantly increased number of metastatic nodules, but not their size, indicating increased tumor cell colonization of the lungs in both models (new results in updated Fig. 7d-i). Thus, our results show that AZD-1208 increased metastatic colonization of the lungs by AZD-1208 insensitive as well as AZD-1208 sensitive tumor cells.

The new results are consistent with previous reports of anti-tumor effects of PIM inhibitors in sensitive cancer cell lines (Santio & Koskinen 2017*). Therefore, our results support the intriguing hypothesis that the effect of PIM inhibition depends on the addition of tumor cells to PIM mediated cell survival, whereas PIM inhibition independently of a tumor type, increases endothelial permeability, potentially explaining the controversial results of PIM inhibition in clinical cancer trials (Malone et al. 2020*).

*Santio, N. M. & Koskinen, P. J. PIM kinases: From survival factors to regulators of cell motility. International Journal of Biochemistry and Cell Biology (2017) DOI: [10.1016/j.biocel.2017.10.016](https://doi.org/10.1016/j.biocel.2017.10.016)

*Malone, T. et al. Current perspectives on targeting PIM kinases to overcome mechanisms of drug resistance and immune evasion in cancer. Pharmacol. Ther. 207, 107454 (2020). DOI: [10.1016/j.pharmthera.2019.107454](https://doi.org/10.1016/j.pharmthera.2019.107454)

Minor points:

1. Fig. 1b: All cluster numbers should be shown.

Author response: We have corrected accordingly and labelled all clusters in all treatments in the updated Fig. 1c of the revised version of our manuscript.

2. Fig. 2c and d: Color legends should be provided.

Author response: We have modified the CellChat data and ended up removing the heatmaps, since the ligand-receptor pair communication strength per each cluster is visible from the bubble blot in the updated Fig. 2b, which also includes the scale for the probability. Furthermore, expression of selected rCap (crEC) ligands in ctrl vs 6h B16-F10 lung is visible from the new feature plots included in Fig. 2d of the revised manuscript.

3. Fig. 6 b-d: The left and right bar graphs should be labeled.

Author response: In the revised manuscript the data from Fig. 6b-d has been transferred to Fig. 5. We have clarified the labelling of these bar graphs in the updated Fig. 5g-l.

Reviewer #2: ECs, vascular biology, metastasis

This is an interesting paper as it has particularly explored the early phase of metastasis and establishment of tumours in the lung, in contrast to many other papers that have looked at established metastases by single-cell sequencing. Intravenous administration of B16-F10 melanoma tumour cells was used for initial experiments and justified because of the need to look at early time points. This showed that after six hours melanoma clusters and single cells were detectable in the lungs, but this dramatically dropped after 12 hours. PECAM1-positive endothelial cells were isolated from the lungs six hours after the injection and analysed by SCS. An additional discriminator was introduced in the analysis, CDH 5 to ensure that endothelial cells were analysed.

Using the previous annotations from other groups showed the expected distribution of endothelial subtypes. However, more detailed analysis showed there were four clusters in the general capillary group. This is a critical point and particularly figures 1B and C should be shown at higher resolution or in larger format. It is not clear how figure 1C fits into 1B. Additionally, there seems to be a small group below 4 on the diagram that is not discussed.

Author response: We thank the reviewer for constructive and positive comments, which have led to significant improvements of our manuscript.

In the revised version of our manuscript, we have included new results from two additional independent scRNA-Seq experiments (biological replicates included as experiments 2 and 3) both consisting of control and metastatic lung samples at 6 hours (two samples) and at 30 hours (one sample) post melanoma injection. All the data was integrated and reanalysed. The new UMAPs have been included in updated Fig. 1c which better represents the critical features of our data.

The scRNASeq datasets are deposited in the GEO repository:
Exp1 (original dataset): GSE235394, access code: unmnswysnbkbfed
Exp2 (new dataset):GSE253749, access code: sdcbkssgblpydyn
Exp3 (new dataset): GSE253751, access code: otclwsysybzuytf

In addition, we provide a new trajectory analysis using RNA velocity to investigate the directionality of the gene expression changes based on ratios of spliced and unspliced mRNAs between control and metastatic samples. The velocity analysis confirms our original results that rCap (crEC) are derived from general capillary ECs and represent the future state of gCap ECs. The new velocity analysis is presented in updated Fig.1g and discussed on p. 6 of the revised manuscript.

In comparison to our previous analysis, we used a slightly lower resolution in separation of the clusters, which does not reveal the gCapV, gCapA and crEC-ori in our original manuscript. Since these clusters have not been validated *in vivo* we consider that this resolution is well justified, and that the analysis of the additional gCap clusters is a subject of further work.

Also, if the same more detailed analysis was done on other groups shown would more subgroups emerge?

Author response: As explained above, we performed a new analysis, with two additional independent experiments, resulting in >40000 ECs. The new analysis with significantly increased EC number robustly identified the rCap (crEC) in the metastatic but also in the healthy lungs. However, further subclustering of the gCaps is a subject of a future study.

Several marker genes for venous capillaries in this paper (Figure.1f) match marker genes of post-capillary venules found in murine LLC lung tumours in this study: Cancer Cell. 2020 Jan 13;37(1):21-

36.e13. doi: [10.1016/j.ccell.2019.12.001](https://doi.org/10.1016/j.ccell.2019.12.001). Is it possible that they are the same EC but differently annotated?

Author response: This is an interesting question. As suggested by the reviewer, we have carefully compared our data to that by Goveia et al. 2020* and included these new results in the Supplementary Fig. 8 in the revised version of our manuscript. The venous capillaries (gCapV, annotated in our original manuscript) did not especially share similarity with markers of the PCVs, however, the rCap (crEC) shared markers (shown in the earlier Fig. 1f, updated version in Fig. 1h) with the PCVs in the vasculature of the LLC tumor, whereas PCVs have not been previously identified in healthy mouse lungs. However, in addition to PCVs, the rCap (crEC) signature is enriched in tumor capillaries and breach ECs, indicating that rCap (crEC) markers are widely expressed in the tumor vasculature, supporting their regulation by tumor cells. These results are discussed in the revised manuscript on p. 9-10 and p. 15.

*Goveia, J. et al. An Integrated Gene Expression Landscape Profiling Approach to Identify Lung Tumor Endothelial Cell Heterogeneity and Angiogenic Candidates. *Cancer Cell* 37, 421 (2020). DOI: [10.1016/j.ccell.2020.03.002](https://doi.org/10.1016/j.ccell.2020.03.002)

On the data and figure 1, INHBB is the most significantly changed and PIM3 is amongst the least significant with a similar fold change, so the explanation for choosing PIM3 needs to be more fully explained. Does INHBB induce PIM3?

Author response: In the revised manuscript we have included two additional scRNASeq data sets and reanalyzed the data, which recapitulated the marker genes of our original analysis, while the order of the markers was slightly altered. The results are now shown based on adjusted P value instead of the earlier Log2FC. In this list, *Pim3* is the top 13th marker right after *Inhbb*.

We have also now better explained our strategy to focus on *Pim3* in our revised manuscript on p. 11. The major reason that made us to focus on *Pim3* was its known role as an oncogene and a drug target, whereas its function in ECs was unknown. Moreover, GSEA identified the JAK/STAT pathway among the strongest significantly upregulated pathways, which is supported by new bulk RNASeq data from cultured HUVECs** (new data in Supplementary Fig. 7). Since the JAK/STAT pathway is a known regulator of metastasis and a known upstream regulator of the *Pim* pathway, it led us to consider the function of *Pim3* upregulation in the metastatic EC niche.

** The new bulk RNASeq data are deposited in the GEO repository GSE251979, under access code utitaigtvobtit.

To test whether *Inhbb* induces *Pim3* mRNA, we measured *Pim3* expression using RT-qPCR in Activin B-treated HUVECs. Interestingly, the results show that Activin-B treatment increases *Pim3* expression, suggesting further reinforcement of *Pim3* expression through multiple signals in the metastatic niche. We have included a figure showing the results (Figure 1), which can be added to the revised manuscript if needed. However, we suggest not including this data in the current manuscript and focusing on it in our future work.

[Figure Redacted]

In Figure 2c missing colour scale for the green heatmaps.

Author response: We have reanalyzed our new data using the CellChat and decided not to visualize the results using heatmaps, since the ligand-receptor communication strength is visible from the bubble blot in the updated Fig. 2b of the revised manuscript, including probability scale.

Figure 2e is not particularly convincing for a difference in the control vs melanoma for VegfA interactions in crEC. Statistical comparison should be carried out to justify the comment. Similarly, in figure 2b, the Vegf axis is lower than in the arterial capillaries with tumour and is more induced in the latter. Please discuss. In Figure 2c Vegf pathway, it looks to me that signaling between 5 and is still stronger than between 4 [crEC] and other pathways. Also, signaling from Notch seems to be go down from 4, not up, with a general reduction. The meaning and code for the lines is not provided.

Author response: Due to the two new scRNA-Seq experiments included in our revised manuscript, we present a new CellChat analysis in the updated Fig. 2b-d. Only statistically significantly upregulated pathways (communication probability $P < 0.01$) are listed in updated Fig. 2b, some of which are visualized using circle plots in updated Fig. 2c and expression of rCap (crEC) ligands using feature plots instead of violin plots in Fig. 2d. We also provide the adjusted P values for the upregulated genes of the pathways mentioned in the Fig. 2b in new Supplementary Table3. The results show upregulation of Vegf and Dll4 signaling via the rCap (crEC). In addition, Coll4a2, Esam and the activin pathway are increased. Altogether, the signaling interactions mediated via rCap (crEC) and other lung EC identities are altered from healthy condition, already during the first 6 hours when tumor cells are increased in the circulation and characterized by previously identified angiogenic pathways (discussed on p. 6-7 and p.15).

In Figure 3 the melanoma cells are shown in green and there seems to be a large distance, relatively several cell diameters, between the melanoma and IINHDB and PIM3 staining. Also, interestingly the cells highlighted by the IINHBB probe seem larger than those with the PIM3 probe, although this may simply be the relative abundance of the RNA. Several representative areas should be presented and more clarity in the explanation of the image, perhaps with arrows. A stain showing the outlines of the cells or PECAM1 protein would help understand the image far better, even if serial sections.

Author response: We have updated Fig. 3 and included additional representative areas of the RNAScope analysis in Supplementary Fig. 5. We provide a co-localization analysis of the coexpression of *Pim3*, with *Inhbb* and *Bcl3* mRNAs in the lung ECs along with higher magnification images (new Fig. 3d-l) to better capture the ISH signal between healthy and metastatic lung. As suggested by the reviewer, we also assume, that *Inhbb* ISH signal abundance but not cell size is likely explanation for larger *Inhbb* ISH area compared to *Pim3*.

In addition, we had quantified the ISH signal in areas $<50 \mu\text{m}$ distance from the cancer cells (Supplementary Fig. 5f in the revised manuscript). This was considered a rough estimate of close localization in 7 μm sections with estimated size of ECs: “The shape of the endothelial cells varies across the vascular tree, but they are generally thin and slightly elongated, their dimension being roughly 50–70 μm long, 10–30 μm wide and 0.1–10 μm thick” (The Endothelium: Part 1: Multiple Functions of the Endothelial Cells—Focus on Endothelium-Derived Vasoactive Mediators).

We agree with the reviewer that the RNA ISH is not optimal for detecting cell borders. However, we were unable to stain PECAM1 after the fixation protocol used for RNAScope. Therefore, we provide new results using Podocalyxin and α SMA staining of serial sections in the updated Supplementary Fig. 5 to better visualize the lung tissue.

Overall, there is a lack of confirmation of the target genes by immunohistochemistry in the metastases.

Author response: We agree with the reviewer that showing the protein levels would be important. However, RNAscope analysis was selected due to the lack of suitable antibodies for IHC analysis. To answer the reviewer's comment, we performed additional Western blots of lysates of ECs isolated from control and metastatic lungs. Updated image of PIM3 protein level has been added to updated Fig. 3m-n. Furthermore, to support our analysis, we provide RNA and protein level analysis of RhoC, an additional rCap (crEC) DE gene (DE gene analysis shown in Fig. 2e) for the reviewer only, although if needed the data can be included in the manuscript.

The *in vitro* work with coculture of melanoma and HUVECs clearly shows the upregulation of PIM3 and INHBB after 16 hours. It would be interesting to know the time course and whether there is greater induction over 24 hours, which may mimic more the duration of the experiments *in vivo*?

[Figure Redacted]

Author response: In accordance with the reviewer's suggestion, we performed a time course analysis of HUVEC + B16-F10 cocultures, which however was compromised due to the overgrowth of tumor cells. Therefore, we used conditioned media from 24 h and 48 h cultured B16-F10 to induce HUVEC for 6, 16 or 24 h, which induced the rCap (crEC) marker genes *PIM3*, *BCL3* and *INHBB*. To simplify the data, we included in the revised manuscript only results using the 48 h conditioned medium for 6 and 16 h stimulations of HUVECs (updated Fig. 4k-m and Supplementary Fig. 6j).

A key question that should be answered by co-culture in separate wells is whether the initial induction of PIM3 was mediated by a soluble factor from the melanoma cells, which might be relevant for many other pathways e.g. interferon secretion. In figure S2, top 3 pathways are all interferon related, so I think the transfer of soluble factors could be a key link, needs to be tested.

Author response: We agree with the reviewer that this is an important point. Therefore, we tested the conditioned media from three different cancer cell lines (B16-F10, LLC, 4T1), which induced rCap (crEC) marker gene expression in HUVEC, indicating that rCap (crEC) markers are upregulated by tumor cell secreted factors. The new data is included in the updated Fig. 4k-m and Supplementary Fig. 6j.

I am not sure how the hypothesis is supported by the human clinical data. Although it shows these genes are upregulated, the hypothesis is that these are transient and involved reversibly in the early niche. This seems unlikely in the studies of the human metastases and should be discussed further.

Author response: This is an interesting point. To best of our knowledge, there is however, no data available from the early premetastatic niche of human cancer, precluding analysis of human data from a setting comparable to our preclinical model.

Nevertheless, to substantiate the potential translational implications of our study, we performed additional analysis using publicly available scRNA-Seq data from the Human Lung Cell Atlas (HLCA, <https://hlca.ds.czbiohub.org/>). Specifically, we analyzed ECs from healthy human lung and from lung tissue adjacent to lung tumor from carcinoma patients. This was justified since rCap (crEC) were found also in the healthy mouse lungs from our new, much larger EC dataset. Interestingly, in addition to previously identified EC identities, the rCap (crEC) cluster was identified in the human lungs, based on

rCap (crEC) marker gene signature (new Supplementary Fig. 9a-e). Furthermore, we integrated the human lung EC dataset with our mouse lung EC dataset, which preserved previously annotated clusters, and confirmed the presence of the rCap (crEC) in the human lung (new Supplementary Fig. 9f-j). We also tested different integration algorithms implemented in Seurat 5, including RPCA and Harmony, and found that the clustering results remained consistent across these methods. These results therefore reveal rCap (crEC) as a novel gCap EC cluster in both mouse and human lungs.

To understand about the temporal regulation of rCap (crEC) we performed scRNA-Seq at 30 h post melanoma injection, in addition to original 6 h time point. This revealed downregulation of rCap (crEC) markers, correlating with decreased melanoma cell numbers at 30 h, when compared to 6 h. Thus, we conclude that rCap (crEC) are upregulated in response to tumor cells, and the decreased tumor cell numbers by 30 h (Supplementary Fig. 1b) led to decreased rCap (crEC) marker expression at this longer timepoint (new results in Fig. 1h, i).

The further work on PIM3 and regulation of endothelial junctions clearly shows the downregulation of CHD5 and confirms the increased vascular leakage and metastasis in the lungs.

This study is important because of its direct impact on studies and clinical research involving PIM 3 inhibitors and the risks. It is interesting though a recent phase one study in hematopoietic neoplasms showed toxicity but no evidence of activity.

We thank the reviewer for the positive comments on our manuscript.

Reviewer #3 (Remarks to the Author):

In this manuscript, Santio et al. characterised the diverse endothelial cell types in the lung metastatic niche after arrival of melanoma cells and revealed a novel cluster of cancer responding ECs. Their scRNA-seq results pointed to the Jak-Stat activated Pim 3 as a marker for these cells. Surprisingly, inhibition of Pim3 increased vascular leakage and metastasis by impairing the EC barrier.

The quality of the data appears very good. The new EC cluster that has been identified could have potential clinical importance. However, I have some questions and comments with regards to the analysis of the data that need to be addressed.

Author response: We wish to thank the reviewer for positive and constructive comments on our manuscript that helped to improve our manuscript.

Major comments:

The authors used very stringent criteria for quality control, very high min of number of genes (1500) and quite low max number of genes (4500), as well as only 3% of mitochondria, was there a reason for such stringent QC metrics?

Author response: We used strict criteria to only include high quality cells in the final data analysis in the original submitted manuscript. However, in the revised version of our manuscript we performed two additional independent biological replicates (experiment 2 and experiment 3, containing samples from control lungs and at 6h and 30h post-melanoma injection), which were included in new analysis. Following the latest Seurat 5 vignette in analysis, we decided to lower the criteria, since the quality of our data in general was very high. In the revised version of the manuscript, we have used less stringent criteria to analyze the original scRNASeq data set, along with the two new data sets (1,000 to 5,000 genes, 20 % of UMIs derived from mitochondrial genome). Prior to filtering the data, we also run doublet detection, and after filtering, removed the non-ECs and reclustered the EC and melanoma subset and performed analysis to the reclustered subset (new Supplementary Fig. 2f-j).

It is noteworthy that due to the very high quality and consistency across independent biological replicates, the results and findings are robust regardless of variations in quality control filtration and thresholds. For example, increasing the mitochondrial percentage threshold from 15% to 20% results in the removal of, on average, around less than 10 additional cells per sample. To more conceptually address the reviewer's concern, we carefully examined the annotated cell clusters for any technical biases. We found no evidence of technical biases related to the number of UMIs, detected genes, mitochondrial percentage, or doublet percentages.

The scRNASeq datasets are deposited in the GEO repository:

Exp1 (original dataset): GSE235394, access code: unmnswysnbkbfed

Exp2 (new dataset): GSE253749, access code: sdbkssgblypdyn

Exp3 (new dataset): GSE253751, access code: otclcwsybzuhtyf

Have the authors performed statistical analysis for the differential abundance between control and metastatic lung? A method such as Milo would be helpful for this.

Author response: We thank the reviewer for raising up this crucial point. As explained above, in the revised manuscript, we performed two additional experiments 6 h post-injection of melanoma cells. Importantly, when the 3 independent scRNA-Seq experiments were analyzed using scSCODA (Büttner et al. 2021*), we found a statistically significant increase in the cell number of the rCap (crEC) cluster. Notably, the capillary EC clusters showed a trend of decreased cell number, but this was not statistically significant. These new results are included in the updated Fig.1f of the revised version of our manuscript.

*Büttner, M., Ostner, J., Müller, C.L. et al. scCODA is a Bayesian model for compositional single-cell data analysis. *Nat Commun.* **12**, 6876 (2021). DOI: [10.1038/s41467-021-27150-6](https://doi.org/10.1038/s41467-021-27150-6)

Fig 1c - the trajectory analysis results are a bit confusing for me, I can't really see a clear trajectory on this plot. Can the authors reproduce their results using a different method? Otherwise the claim that the novel EC cluster is derived from venous capillaries would be very speculative.

Author response: We agree that the Slingshot analysis is not optimal, even though we used an automatic selection of the starting cluster. Therefore, according to the reviewer's suggestion, we performed RNA velocity analysis, which shows the directionality of the gene expression changes based on ratios of spliced and unspliced mRNAs. The velocity analysis supports our original results that rCap (crEC) is derived from general capillary ECs and represent the future state of gCap. The new velocity analysis is presented in updated Fig.1g and discussed on p. 6 of the revised manuscript. In the revised manuscript we have also interpreted the data more cautiously by combining the gCapA and gCapV clusters of our original manuscript into a single gCap cluster, as validation of these subclusters was not in the scope of this manuscript, although it will be of interest in the future.

Fig 2e - is there a statistically significant difference in expression of the selected genes?

Author response: Instead of the violin plot in original Fig. 2e, we now provide the adjusted P values for the fold change of the communication pathways identified using CellChat in the Supplementary Table 2 and we highlight the statistically significant differences between control and metastatic lungs for the main rCap (crEC) secreted factors in feature plots shown in new Fig. 2d.

Why was a UMAP dimensionality reduction used for CellChat, instead of cluster labels or a more appropriate dimensionality reduction for clustering such as PCA?

Author response: The selection to use the Seurat 5 object after UMAP dimensional reduction was based on <https://github.com/jinworks/CellChat> instructions.

The authors performed cell-cell communication analysis using only the sequenced EC data. It would be much more relevant to try this with cancer and other cells, using publicly available data, in order to see how the cancer cells potentially communicate with the cancer responding ECs.

Author response: We completely agree with the reviewer that cell-cell communication analysis among heterotypic cells in the metastatic niche would be of great interest. However, the short time point at 6 h post melanoma injection does not support the involvement of multiple transcriptionally and translationally regulated signaling cascades between heterotypic cells in the lungs. Instead, our experimental set-up is best suited to investigate the immediate effects of tumor cells on the ECs. Moreover, at this early time point, inflammatory cell numbers were not altered in the healthy vs metastatic lungs in the B16-F10 model (as shown in Supplementary Fig.3 in the revised manuscript).

To answer the reviewer's question, we performed additional scRNA-Seq analysis where we collected both ECs and cancer cells. Although we increased the number of injected cancer cells, the number of cancer cells remained low (91). CellChat analysis revealed melanoma-EC communication pathways as potential candidates regulating rCap (crEC). However, the analysis of their role on rCap (crEC) will be a subject of further study. This data is presented in new Supplementary Fig. 4.

Fig S3c, was there any statistically significant difference in expression of the selected genes?

Author response: After including two new scRNA-Seq experiments at 6 h post-injection and one experiment 30 h post-injection, we provide new results showing a statistically significant increase in the expression of *Pim3* and *Pim1* in rCap (crEC) as compared to other cell types, while also veins express relatively high levels of *Pim1* and *Pim3*, whereas *Pim2* is not expressed by ECs (Supplementary Fig. 10a). Furthermore, after melanoma injection both *Pim1* and *Pim3* FC are statistically significantly

upregulated in the melanoma sample at the 6 h time point as compared to the control lung (violin plot at Fig. 1l and dotplot at Supplementary Fig.10b).

From the analysis of the publicly available data, it seems that the specific markers of the cancer responding ECs are also present in endothelial cells of primary tumours, have you checked this and what is your comment on that?

Author response: To answer this question, we performed two separate analyses in the revised manuscript. First, we compared our rCap (crEC) gene signature from early metastatic lungs to previously published scRNA-Seq data by Goveia et al. 2020* from mouse lung tumor ECs (TEC) derived from orthotopic Lewis Lung Carcinoma (Lung Tumor ECTax, Mouse Lung (https://endotheliomics.shinyapps.io/lung_ectax/)). The tumor EC dataset was refiltered and normalized similarly to our data and indeed, the rCap (crEC) marker genes are upregulated in the TECs, especially in postcapillary venules, tumor capillaries and breach cells. We conclude that tumor cells induce upregulation of rCap (crEC) markers and a gene signature, with similarity to angiogenic tumor vasculature.

*Goveia, J. et al. An Integrated Gene Expression Landscape Profiling Approach to Identify Lung Tumor Endothelial Cell Heterogeneity and Angiogenic Candidates. *Cancer Cell* 37, 421 (2020). DOI: [10.1016/j.ccell.2020.03.002](https://doi.org/10.1016/j.ccell.2020.03.002)

Second, we chose to investigate the presence of rCap (crEC) in healthy human lungs. This was justified since rCap (crEC) were found also in the healthy mouse lungs from our new, much larger scRNA-Seq dataset. Specifically, we analyzed the EC subset from the Human Lung Cell Atlas (HLCA, <https://hlca.ds.czbiohub.org/>) including ECs from healthy human lung and from lung tissue adjacent to lung tumor from carcinoma patients. Interestingly, in addition to previously identified EC identities, the rCap (crEC) cluster was identified in the human lungs, based on rCap (crEC) marker gene signature (new Supplementary Fig. 9a-e). Furthermore, we integrated the human lung EC dataset with our mouse lung EC dataset, which preserved previously annotated clusters, and confirmed the presence of the rCap (crEC) in the human lung (new Supplementary Fig. 9f-j). We also tested different integration algorithms implemented in Seurat 5, including RPCA and Harmony, and found that the clustering results remained consistent across these methods. These results therefore reveal rCap (crEC) as a novel gCap EC cluster in both mouse and human lungs, with enriched marker gene expression in several distinct mouse tumor EC clusters.

Minor:

Fig 1g, 2a, it would be visually better if the annotations of clusters are used instead of numbers.

Author response: We have now used cluster names instead on numbers.

Changes in the main figures:

Figure 1:

- a. New experimental outline for the three biological replicates for scRNA-Seq.
- b. Original data, labeled as 1b.
- c. New UMAP consisting of three biological replicates (n=3 independent experiments) integrated for PBS Ctrl and B16-F10 6h and an additional 30 h time point in experiment 3.
- d. New feature plots for the most prominent markers, replaces dot plot in the original manuscript. Top 50 marker list for each cluster is available in Supplementary Table 1.
- e. New violin plots for the most prominent cluster markers.
- f. New statistical analysis of cell composition using scCODA, demonstrating statistically significant increase of rCap cluster, replaces cell numbers in the original manuscript.
- g. New RNA velocity analysis, replaces slingshot trajectory analysis (previous Fig. 1c).
- h. New dotplot representation for the rCap marker FC between control and metastatic samples, replaces original graph and heatmap for rCap markers (Fig. 1f-g).
- i. New violin plot showing the rCap marker expression between ctrl, 6h and 30 h treatments.
- j. Original results, except one marker gene has been removed.

Figure 2:

- a. New cell communication heatmap due to addition of two new scRNA-Seq experiments.
- b. New cell communication bubble plot due to addition of two new scRNA-Seq experiments.
- c. New Fig. 2c combining data from original Fig. 2c-d, heatmaps were removed and one additional pathway was identified.
- d. New Fig. 2d shows feature plots for selected pathways, replaces violin plots in original Fig. 2e.
- e. The identification of the rCap cluster in both control and metastatic lungs enabled DE gene analysis between control and metastatic lungs, shown in a new heatmap (replaces the rCap marker-mediated pathways of original figure version).
- f. New GSEA of rCaps in 6h metastatic vs control lungs (replaces original KEGG pathways in Fig. 2g)
- g. New dot plot of JAK-STAT pathway FC values in 6h metastatic vs control lungs (replaces gene set, heatmap and feature plots in original Fig. 2h-j).
- h. Overview to Jak-Stat pathway has been added.

Figure 3:

- a-c. New feature plots of rCap marker expression in the scRNA-Seq data, this supports the selection of the markers for RNAscope analysis.
- d-e. New representative RNAScope image panel (replaces original Fig. 3a), new rCap marker *Bcl3* has been included.
- f. New quantification RNAScope signal area normalized to nuclear area (previously Fig. 3c). Original Fig. 3c quantification has been transferred to the Supplementary Figure S5f.
- i, k. New representative magnified images showing the rCap marker *Pim3+Bcl3* and *Pim3+Inhbb* coexpression in *Pecam1+* ECs. (Fig. 3d has been moved to Supplementary Fig. 10 f)
- j-l. New quantification of the coexpression.
- m-n. Additional samples have been analyzed for PIM3 using Western blotting (original Fig. 3e).

Figure 4:

- a. No major changes
- b. Original data, relabeled as 4b.
- c. New RT-qPCR for *Bcl3* has been added showing its upregulation in lung ECs during s.c. melanoma metastasis.

- d. Small adjustment to the time scale up to 27d.
- e. Original data, relabeled as 4e.
- f. New RT-qPCR for *Bcl3* has been added showing its upregulation in lung ECs during orthotopic 4T1 metastasis.
- g. Original data, relabeled as 4g.
- h. New RT-qPCR for *Bcl3* has been added showing its upregulation in liver ECs during orthotopic 4T1 metastasis.
- i. Original schematic of HUVEC treatment, relabeled as 4i.
- j. Original Fig. 4h, relabeled.
- k. Schematics of HUVEC treatment.
- l. New RT-qPCR analysis showing rCap marker upregulation in HUVECs treated with B16-F10 cell conditioned-medium.
- m. New RT-qPCR analysis showing rCap marker upregulation in HUVECs treated with 4T1 and LLC cell conditioned-medium.

Figure 5:

Experiments showing the effect of PIM3 shRNA on cadherin 5 and catenins in cultured ECs from original fig. 5 and fig. 6 have been combined in one new fig. 5

Figure 6:

Experiments showing the effect of PIM inhibitor on EC barrier function and cadherin 5 *in vitro* and *in vivo* from original fig. 5 and fig. 8 have been combined in new fig. 6.

Figure 7.

New data using 4T1 and LLC metastasis models have been added in Fig. 7d-i.

Figure 8.

Contains only the graphical illustration, which has been modified from the earlier version.

Changes in supplementary figures and tables:

Supplementary Fig. 1

Additional data points have been added to Supplementary Fig. 1a and Fig. 1b.

Supplementary Fig. 2

New Supplementary Fig. 2 (replaces original Fig. 2) contains additional information related to cell sorting, and quality controls and methodological data for scRNA-Seq.

Supplementary Fig. 3

No changes.

Supplementary Fig. 4

New results of melanoma – EC communication using CellChat, based on new scRNASeq experiments including melanoma cells.

Supplementary Fig. 5

New results, containing podocalyxin IHC of a serial lung section to better visualize the lung vasculature as well as panels and quantification demonstrating rCap marker *Bcl3* expression in the vicinity of melanoma cells in the lungs.

Supplementary Fig. 6

New results in Supplementary Fig. 6j showing that B16-F10 conditioned medium i.e. melanoma cell - secreted factors induce the upregulation of rCap markers, time point 16h. 6h time point presented in Fig. 4l-m. Supplementary Fig. 6h-i are original results from previous Fig. 4e-g and Supplementary Fig. 6a-g are original results from Supplementary Fig. 4a-g.

Supplementary Fig. 7

New results from bulk RNA-Seq from melanoma conditioned-medium-treated HUVECs.

Supplementary Fig. 8

New results analysing rCap markers in mouse lung tumor ECs.

Supplementary Fig. 9

New results including analysis and visualization of rCap cluster in the human lungs. This replaces the previous Supplementary Fig. 6.

Supplementary Fig. 10

Updated and combined from original Supplementary Fig. 3 and Supplementary Fig. 7.

Supplementary Fig. 11

Updated from original Supplementary Fig. 8.

Supplementary Fig. 12

New figure demonstrating e.g. the effect of PIM inhibitor on leakage in 4T1 orthotopic mammary carcinoma model, and additional supplementary data for main Fig. 7.

REVIEWERS' COMMENTS

Reviewer #1 (Remarks to the Author):

The authors have satisfactorily addressed my comments.

Reviewer #2 (Remarks to the Author):

The authors have answered my queries well. In each case, further experimental work has been done to enhance statistical strength or follow my suggestions. The substantial further work makes the findings secure, emphasises their discovery on PIM3 kinase and its relevance to cancer therapy and biology.

Reviewer #3 (Remarks to the Author):

The authors have addressed all of my comments and the manuscript has been significantly improved.

Reviewer #3 (Remarks on code availability):

The code was easy to understand, with detailed comments throughout it and should enable reproducing the results.